# ENHANCING VISUAL REPRESENTATIONS FOR EFFICIENT OBJECT RECOGNITION DURING ONLINE DISTILLATION

## ABSTRACT

We propose ENVISE, an online distillation framework that ENhances VISual representations for Efficient object recognition. We are motivated by the observation that in many real-world scenarios, the probability of occurrence of all classes is not the same and only a subset of classes occur frequently. Exploiting this fact, we aim to reduce the computations of our framework by employing a binary student network (BSN) to learn the frequently occurring classes using the pseudo-labels generated by the teacher network (TN) on an unlabeled image stream. To maintain overall accuracy, the BSN must also accurately determine when a rare (or unknown) class is present in the image stream so that the TN can be used in such cases. To achieve this, we propose an *attention triplet loss* which ensures that the BSN emphasizes the same semantically meaningful regions of the image as the TN. When the prior class probabilities in the image stream vary, we demonstrate that the BSN adapts to the TN faster than the real-valued student network. We also introduce *Gain in Efficiency* (GiE), a new metric which estimates the relative reduction in FLOPS based on the number of times the BSN and TN are used to process the image stream. We benchmark CIFAR-100 and tiny-imagenet datasets by creating meaningful inlier (frequent) and outlier (rare) class pairs that mimic real-world scenarios. We show that ENVISE outperforms state-of-the-art (SOTA) outlier detection methods in terms of GiE, and also achieves greater separation between inlier and outlier classes in the feature space.

## 1 INTRODUCTION

Deep CNNs that are widely used for image classification (Huang et al. (2017)) often require large computing resources and process each image with high computational complexity (Livni et al. (2014)). In real-world scenarios, the prior probability of occurrence of individual classes in an image stream is often unknown and varies with the deployed environment. For example, in a zoo, the image stream input to the deep CNN will mostly consist of *animals*, while that of *vehicles* would be rare. Other object classes such as *furniture* and *aircraft* would be absent. Therefore, only a subset of the many classes known to a deep CNN may be presented to it for classification during its deployment.

To adapt to the varying prior class probability in the deployed scenario with high efficiency, we propose an online distillation framework - ENVISE. Here, we employ a high capacity general purpose image classifier as the *teacher network* (TN), while the *student network* (SN) is a low capacity network. For greater efficiency and faster convergence, we require the coefficients of the SN to be binary and refer to it as the binary student network (BSN). When the BSN is first deployed, it is trained on the unlabeled image stream using the predicted labels of the TN as *pseudo-labels*. Once the BSN converges to the performance of the TN, it is used as the primary classifier to rapidly classify the frequent classes faster than the TN. However, if a rare class appears (i.e. a class absent during online training) in the image stream , the BSN must accurately detect it as a class it has not yet encountered, which is then processed by the TN. Since the BSN is trained only on the frequent classes, we refer to these classes as *inlier* (IL) and the rare classes as *outlier* (OL). It is important to note, that the OL classes are *outliers* with respect to the BSN only, but are known to the TN. Detecting extremely rare classes which are unknown to both the BSN and TN (global unknowns) is beyond the scope of this paper. Thus, assuming that the TN knows all possible classes from the deployed environment, we

aim to increase the overall efficiency of the system (without sacrificing performance) by exploiting the higher probability of occurrence of frequent classes in a given scenario.

Our approach for detecting OL classes is motivated from the observation by Ren et al. (2019), where networks incorrectly learn to emphasize the background rather than the semantically important regions of the image leading to poor understanding of the IL classes. We know that *attention maps* highlight the regions of the image responsible for the classifier's prediction (Selvaraju et al. (2017)). We empirically observe that, the attention map of the BSN may focus on the background even when the attention map of the TN emphasizes the semantically meaningful regions of the image. In doing so, the BSN memorizes the labels of the TN, making it difficult to differentiate between the representations of IL and OL classes. To mitigate these issues, we propose an *attention triplet loss* that achieves two key objectives - a) guide the attention map of the correct prediction of the BSN to focus on the semantically meaningful regions, and b) simultaneously ensure that the attention map from the correct and incorrect predictions of the BSN are dissimilar. We show that by focusing on the semantically relevant regions of the image, the BSN will learn to distinguish between the representations of IL and OL classes, thereby improving its ability to detect OL classes.

To assess the overall gain in efficiency of ENVISE, we propose a new evaluation metric - *GiE*, based on the number of times the BSN and TN are used to process the image stream. Since the deployed scene is comprised mostly of IL classes with few OL classes, we expect the BSN to be employed most of the time for classifying IL classes. The TN is used rarely i.e. only when the BSN detects an OL class. We refer to efficiency as the overall reduction in FLOPs in the online distillation framework to process the varying prior probability of classes in the image stream. This term differs from conventional model compression techniques (Frankle & Carbin (2019); Chen et al. (2020)) which process the image stream comprising of classes with equal probability using a single compressed model. To the best of our knowledge, we are the first to propose supervision on attention maps for OL detection, and a new evaluation metric that measures the gain in computational efficiency of an online distillation framework. A summary of our main contributions is:

- **Faster convergence of BSN:** We theoretically justify and empirically illustrate that the BSN adapts to the performance of the TN faster than the real-valued SN (RvSN). We also demonstrate the faster convergence of the BSN for different BSN architectures over its corresponding RvSN.
- **Attention triplet loss** ($L_{at}$), which guides the BSN to focus on the semantically meaningful regions of the image, thereby improving OL detection.
- **New evaluation metric - GiE** to measure the overall gain in computational efficiency of the online distillation framework, and
- **We benchmark CIFAR-100 and tiny-imagenet datasets** with SOTA OL detection methods by creating meaningful IL and OL class pairs. ENVISE outperforms these baseline methods for OL detection, improves separation of IL and OL classes in the feature space, and yields the highest gain in computational efficiency.

## 2 RELATED WORK

**Online distillation :** Distilling knowledge to train a low-capacity student network from a high-capacity teacher network has been proposed as part of model compression (Hinton et al. (2015)). Wang & Yoon (2020) provide a detailed review of different knowledge distillation methods. Mullapudi et al. (2019) propose an online distillation framework for semantic segmentation in videos, while Abolghasemi et al. (2019) use knowledge distillation to augment a visuomotor policy from visual attention. Lin et al. (2019) propose an ensemble student network that recursively learns from the teacher network in closed loop manner while Kim et al. (2019) use a feature fusion module to distill knowledge. Gao et al. (2019) propose online mutual learning and Cioppa et al. (2019) propose to periodically update weights to train an ensemble of student networks. These ensemble based methods require large computational resources and are expensive to train. However, ENVISE involves training a compact model that mimics the performance of the TN with less computation.

**Outlier detection :** Outlier detection or out-of-distribution detection (OOD) refers to detecting a sample from an unknown class (Hendrycks & Gimpel (2016)). Existing SOTA OOD methods use outlier samples during training or validation. Yu & Aizawa (2019) increase the distance between IL and OL samples during training while Vyas et al. (2018); Lee et al. (2018) add perturbations

onto test images and train their network to differentiate between IL and OL samples. However in ENVISE, the BSN is adaptively trained only on IL class images without any knowledge of OL class images, which is a more realistic scenario. Without using outlier classes during training, Hendrycks & Gimpel (2016); Papadopoulos et al. (2019); Bendale & Boult (2016); Hendrycks et al. (2019) employ class probabilities or measure the similarity between IL and OL features. Yoshihashi et al. (2019) generate discriminative feature space using reconstruction based approaches. However, the likelihood of correct classification by these networks is based on background regions as described in Ren et al. (2019). Hence, these methods fail when applied to real-world IL and OL class pairs since the confidence of prediction is based on background information. We overcome this drawback by proposing supervision on attention maps where we guide the BSN to focus on the important semantic regions using the attention map of the teacher network as a reference. This enables ENVISE to classify IL images with high confidence while also improving OL detection.

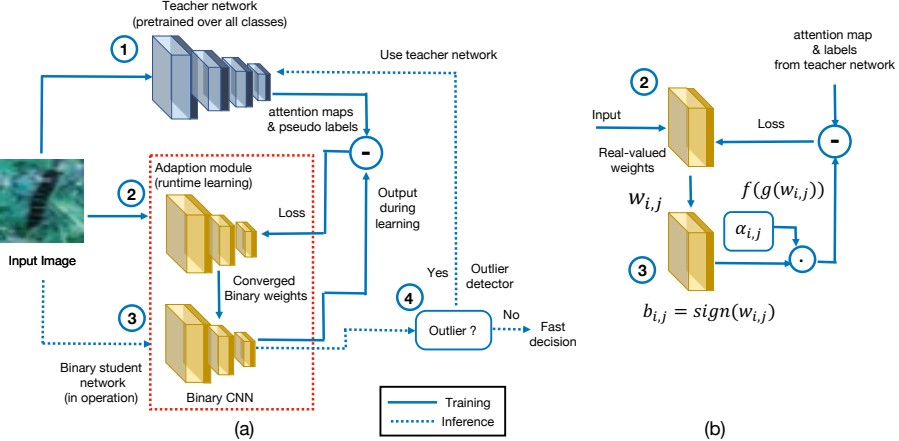

Figure 1: (a) The ENVISE architecture adaptively trains the BSN from predictions of the TN, (b) The process updates real valued weights that minimize the error produced by their binarized version.

## 3 ONLINE DISTILLATION FOR EFFICIENT OBJECT RECOGNITION

The framework of ENVISE is shown in Figure 1 (a). Initially, the TN (indicated by "1" in the figure) classifies the images from the input stream with high accuracy, albeit at a slower rate. The BSN (red dotted box) comprises of an adaptation module ("2") and a binary CNN ("3"). Given an image stream, the adaptation module learns the optimum binary weights to mimic the performance of the TN. Once its accuracy converges to that of the TN, the adaption of the real-valued weights stops, but the inference with binary weights continues. The OL detector ("4") uses the softmax output of the BSN as confidence to either generate the final prediction (if the confidence is high), or treats the image as an OL class and redirects it to the TN for classification.

**Binarization algorithm for student network:** During adaptive training, the BSN does not have access to the labeled data and mimics the behavior of the TN on the IL classes as shown in Figure 1(b). For an input image $x$, let $f(g(w_{i,j}), x)$ represent the output of the BSN where $w_{i,j}$ are real-valued weights of $i^{th}$ kernel in the $j^{th}$ layer, and $g(.)$ is a binarizing function for these weights. During adaptive training, the error between the output of the BSN and TN is minimized by optimizing $w_{i,j}^* = \arg\min_{g(w_{i,j})} \|L\|$, where $L$ is the overall loss function used for training the BSN. While the intermediate weight $w_{i,j}$ is computed during adaptive learning, the binary versions of these weights $w_{i,j}^* = g(w_{i,j})$ are used for fast computation. *After convergence, the real valued weights are discarded, and only the binary weights are used.* The question becomes what is a good choice for the binarization function $g(.)$? Following Rastegari et al. (2016), we find a binary vector $b_{i,j}$ (whose elements are either $+1$ or $-1$), and a non-negative scalar $\alpha_{i,j}$ such that $w_{i,j}^* = \alpha_{i,j}.b_{i,j}$ is the best approximation of $w_{i,j}$ in a minimum squared error sense, where $\alpha_{i,j} = \frac{1}{N}|w_{i,j}|_1$, $b_{i,j} = sign(w_{i,j})$. However, unlike Rastegari et al. (2016) where a standalone network is binarized, our BSN learns the $w_{i,j}^*$ during adaptive training in an online distillation framework. In doing so, the BSN compensates for the effects of binarization on the overall loss function.

Our motivation for using the BSN is that it classifies the image stream with high accuracy and utilizes fewer FLOPs as compared to the RvSN. This results in an increase in efficiency of the overall framework. Furthermore, the BSN converges to the performance of the TN faster than the RvSN, which we theoretically emphasize in the Lemma below and experimentally validate in Section 4. We provide the sketch proof here and the complete proof in Appendix A.1.

**Lemma 3.1.** *Let RvSN and BSN represent the real-valued student network and binary student network respectively with the same network architecture. Let $R(.)$ denote the rate of convergence in terms of accuracy when the student network is adaptively trained using the pseudo-labels from the teacher network. Then, $R(BSN) > R(RvSN)$ for the same image stream and number of iterations.*

***Proof :*** We assume our image stream as $x(n)$ where $n = 1, 2, 3...N$, comprises of $N$ samples from the deployed scenario. Since the weights of the BSN are derived from the RvSN, we prove this lemma first for RvSN and then extend it to the BSN. Let $w^*$ be the optimal weight value which represents network convergence i.e. misclassification error $= 0$.

The weight update rule using back propagation is given by:

$$w(n + 1) = w(n) + \eta x(n)$$
$$[\text{ Since } \eta = 1] \quad w(n + 1) = w(n) + x(n) \tag{1}$$

Computing the bounds of the weights of the RvSN, we have

$$\frac{n^2 \alpha^2}{\|w^*\|^2} \leq \|w(n+1)\|^2 \leq n\beta \tag{2}$$

where $\alpha = \operatorname{argmin}_{x(n) \in C} w^\star x(n)$ and $\beta = \operatorname{argmax}_{x(n) \in C} \|x(k)\|^2$. From eq. 2, we can say that for this inequality to be satisfied, there exists $n_r^\star$ which denotes the optimal number of samples for which the RvSN converges i.e. we obtain $w^\star$ at $n = n_r^\star$. This is given as follows:

$$\frac{n_r^{\star 2} \alpha^2}{\|w^\star\|^2} = n_r^\star \beta$$
$$n_r^\star = \frac{\beta \|w^\star\|^2}{\alpha^2} \tag{3}$$

Thus from eq. 3, we can say that the RvSN achieves convergence after being trained on $\frac{\beta \|w^*\|^2}{\alpha^2}$ number of samples. Since, the weights of the BSN are derived from the RvSN, we substitute the value of binary weight $\hat{w}^*$ from Rastegari et al. (2016) to obtain

$$n_b^\star = \frac{n_r^\star}{N} \tag{4}$$

Here, $n_b^\star$ is the optimal number of samples for which the BSN converges. Comparing eq. 3 and eq. 4, we observe that number of $n_b^* < n_r^\star$ i.e. the BSN takes fewer samples to converge to the performance of the TN than RvSN given the same network architecture.

**Knowledge transfer from teacher to student network**: Given an unlabeled image stream, the BSN is trained from the predictions of the TN as hard pseudo-labels using cross-entropy loss, as $L_d = -\frac{1}{N} \sum_i y_i log(s_i)$. Here, $y$ and $s$ are the pseudo-labels generated by the TN and the softmax output of the BSN respectively, $N$ is the total images in the image stream. Once the BSN converges to the performance of the TN, we employ the BSN as the primary classifier during inference. When the deployed scenario does not change for a long duration, the BSN classifies the IL classes faster than the TN without relying on the latter. This improves the overall efficiency of the framework as the TN is now used to classify only the OL classes (number of OL images $<<$ number of IL images). However, to maintain overall accuracy, the BSN must also accurately determine when the image from the input stream belongs to an OL class so that the TN can be used in such cases.

We know that attention maps highlight the regions of the image responsible for the classifier's prediction (Li et al. (2018)). Some examples of the attention maps from the correct predictions of the BSN and the TN are shown in Figure 2 (a). The first three columns in Figure 2 (a) show that although the BSN (adaptively trained using $L_d$) and the TN both correctly classify the images, their attention maps are significantly different. The attention map from the correct prediction of the BSN

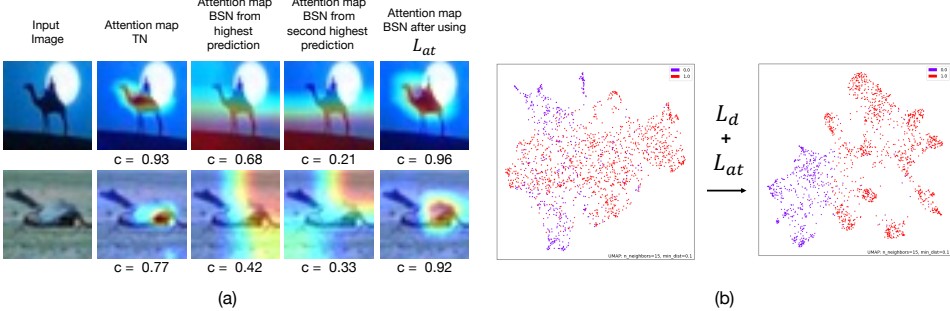

Figure 2: (a) The attention map visualization illustrating that after the BSN is trained using the proposed attention triplet loss, it learns to focus on the semantically meaningful regions in the image. The confidence of prediction ($c$) also increases. (b) Separation between IL and OL classes in feature space improves with the proposed attention triplet loss. More visualizations in Appendix A.4

focuses on the background while that of the TN lies on the semantically meaningful regions of the image. Furthermore, the attention maps of the BSN's correct and incorrect predictions are visually similar which causes the BSN to learn incorrect representations of the IL images while correctly classifying them. To better differentiate between IL and OL classes, the BSN should not memorize the predictions of the TN, but learn proper representations of the IL images.

To achieve this, we propose the *attention triplet loss* given in Equation 5 that causes the BSN's attention map to be similar to that of the TN while forcing the attention map from the BSN's correct and incorrect prediction to be different. We use Grad-CAM (Selvaraju et al. (2017)) to compute the attention map $A$ from the one-hot vector of the predicted class. To generate the attention map triplets, we use the attention map from the TN's prediction $A_t$ as the *anchor*. The *hard positive* attention map $A_{s_p}$ is obtained from correct prediction of the BSN. When the BSN misclassifies an image (prediction of BSN and TN are different), we use the label from the TN to compute $A_{s_p}$. Motivated by the findings in Wang et al. (2019), we observe that the attention maps generated from the correct and incorrect predictions of the BSN are visually similar. Hence, we use the attention map from an incorrect prediction as the *hard negative* attention map $A_{s_n}$ to enforce its separability from $A_{s_p}$. To avoid bad local minima and a collapsed model early on during training, mining suitable hard negatives are important (Schroff et al. (2015)). Hence, we use the attention map from the second most probable class (incorrect prediction) as our hard negative due to its similarity with the hard positive (third and fourth column in Figure 2 (a)). Thus, we formulate the attention triplet loss $L_{at}$ as:

$$L_{at} = \frac{1}{N}\frac{1}{K}\sum_{n}^{N}(\sum_{k}^{K}\left\|A_{t_k} - A_{s_{p_k}}\right\|_2 - \sum_{k}^{K}\left\|A_{t_k} - A_{s_{n_k}}\right\|_2 + \delta) \tag{5}$$

Initially, the hard negative lies within the margin since its squared distance with the hard positive is small. Hence, $L_{at}$ enforces separation between hard positive and hard negative by a distance greater than the value of margin $\delta$, which we empirically set as 1.5. $N$ is the total number of samples and $K$ is the total number of pixels in $A$. Using $L_{at}$ and $L_d$, we formulate our final objective function as:

$$L_f = \lambda_1 L_d + \lambda_2 L_{at} \tag{6}$$

where $\lambda_1$ and $\lambda_2$ are the weights of $L_d$ and $L_{at}$ which we empirically set as 1 and 0.2 respectively. The effect of $L_{at}$ is shown in the last column of Figure 2 (a) where the BSN learns proper representations of the IL image due to its attention map being very similar to that of the TN (second column). Furthermore, the improvement in feature space separation between the IL and OL classes in Figure 2 (b) shows that the BSN not only forms clusters of the different IL classes (in red), but is also well separated from the OL classes (in purple).

## 4 EXPERIMENTAL DETAILS

**Datasets and implementation:** We evaluate ENVISE on CIFAR-100 (Krizhevsky & Hinton (2009)) and tiny-imagenet (TI) (Yao & Miller (2015)) datasets by creating meaningful IL and OL *super-class* pairs that mimic real-world scenarios. On the test set of CIFAR-100 and validation set of TI dataset, we create 12 and 10 super-classes respectively. We summarize our experimental settings in Table 1.

We use DenseNet-201 (Huang et al. (2017)) as the TN, pre-trained on the training set of all classes of CIFAR-100 or the TI dataset individually. The BSN is a VGG16 (Simonyan & Zisserman (2014)) network whose weights, except for the first and last convolution layer, are binarized (Rastegari et al. (2016)). We also compare this with other choices for the BSN including AlexNet, ResNet-18 and ResNet-50. In the Appendix A.3, we illustrate that ENVISE is insensitive to the specific TN and BSN architecture used and achieves high performance gains even with different network architectures.

Table 1: Different pairs of IL and OL super-classes on the CIFAR-100 and TI datasets.

| CIFAR-100 (Krizhevsky & Hinton (2009)) | | | | | | tiny-imagenet (Yao & Miller (2015)) | | | | | |
|---|---|---|---|---|---|---|---|---|---|---|---|
| IL \ OL super-classes | # IL train | # OL train | # IL test | # OL test | # total classes | IL \ outlier super-classes | # IL train | # OL train | # IL test | # OL test | # total classes |
| aquatic animals \ food container ($C_1$) | 840 | 0 | 1200 | 500 | 17 | animals \ garments ($T_1$) | 875 | 0 | 1250 | 900 | 43 |
| flora \ electrical items ($C_2$) | 630 | 0 | 900 | 500 | 14 | reptiles \ edible items ($T_2$) | 630 | 0 | 900 | 800 | 34 |
| fruits \ furniture ($C_3$) | 350 | 0 | 500 | 500 | 10 | aquatic animals \ birds ($T_3$) | 420 | 0 | 600 | 250 | 17 |
| insects \ manmade things ($C_4$) | 560 | 0 | 800 | 500 | 13 | household items \ nature ($T_4$) | 1432 | 0 | 2050 | 700 | 55 |
| animals \ people ($C_5$) | 1750 | 0 | 2500 | 500 | 30 | vehicles \ miscellaneous ($T_5$) | 525 | 0 | 750 | 1800 | 51 |
| outdoor places \ vehicles ($C_6$) | 420 | 0 | 600 | 1000 | 16 | - | - | - | - | - | - |

**Training and evaluation:** Throughout our experiments, we fix the TN and do not train it. We train the BSN using the pseudo-labels and attention map from the TN using the cost function in eq. 6 only on the IL classes of the super-class from Table 1. This is done using a learning rate of $1e^{-4}$ for 10 epochs using the Adam optimizer (Kingma & Ba (2014)) with a batch size of 10. Once the BSN converges to the performance of the TN, we employ it as our primary classifier during inference. To create a more realistic scenario during inference, we random center crop and randomly rotate the image stream between $[-15°, 15°]$. We assume that the distribution of classes will not change rapidly in the deployed scenario, and that the BSN can be used for inference for long durations (e.g days, weeks or months). Thus, the epochs for online distillation are expected to require a small fraction of that time. For each image from the input stream during inference (comprising of IL and OL classes), following Hendrycks & Gimpel (2016), we compute the confidence of prediction from the softmax probability of the predicted class. If the confidence is low, we treat the image as an OL class and transfer it to the TN for classification.

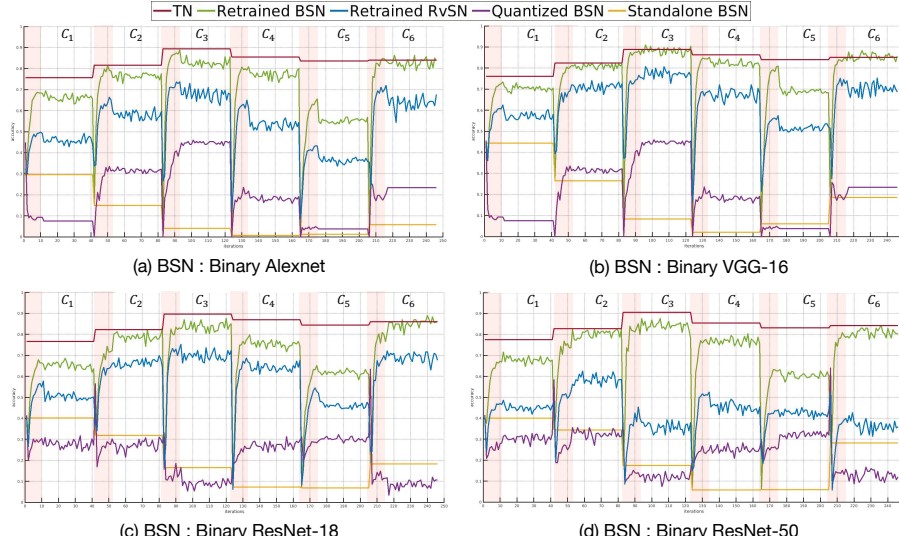

(a) BSN : Binary Alexnet

(b) BSN : Binary VGG-16

(c) BSN : Binary ResNet-18

(d) BSN : Binary ResNet-50

Figure 3: (a) : (d) The BSN converges to the performance of the TN faster than the RvSN and the variants of BSN. The red period illustrates adaptive training and white period illustrates inference. We observe a similar convergence pattern with different binary network architectures.

**Gradually changing the deployed scenario:** When the prior probabilities of the classes change in the deployed scenario, the BSN quickly re-trains to learn the new IL classes and regains efficiency on the new image stream. Figure 3 illustrates the learning behavior of the BSN when the IL and OL classes are changed. Initially, we assume that the image stream is comprised of only IL classes from $C_1$. Once the BSN (in green) converges to the performance of the TN (in red), we stop training the BSN and observe that it achieves an accuracy comparable to the TN's accuracy. The learning behaviour of the RvSN (in blue) is slower than the BSN and has poorer accuracy after 10 epochs.

Directly binarizing the RvSN (purple line) is also worse than the BSN, and substantially differs from the binarization algorithm in Section 3 which compensates for the quantization effect. The yellow line shows the performance of the standalone BSN which is not trained during the adaptive training.

In Figure 3 we keep the IL/OL superclasses the same for 40 epochs, and then switch the input stream to a different scenario. The 10 epochs shaded in red indicate the adaptive training phase, while the unshaded (white) intervals indicate the 30 epochs for inference. We randomly choose our inference phase as 30 epochs. We show in Appendix A.3 that during inference, the combined accuracy of ENVISE is identical to that of the stand-alone TN, which illustrates that ENVISE maintains the overall accuracy of the system. When the scenario changes after 40 epochs (e.g from $C_1$ to $C_2$), we observe similar and consistent learning behaviour, and the BSN retrains quickly from the TN to regain efficiency. We also observe a similar convergence pattern for different BSN architectures when adaptively trained from the same TN in Figure 3. Here, we observe that the rate of convergence for Binary AlexNet is the fastest and that of Binary Resnet-50 is the slowest. This illustrates that a smaller binary network would make an ideal BSN in real-world scenarios, since it would utilize fewer epochs during adaptive training.

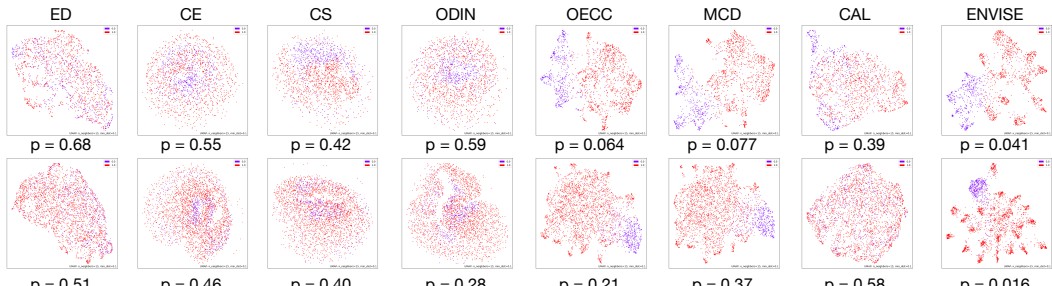

Figure 4: Feature space separation of the IL and OL classes for different baseline methods. $p-$value denotes the overlap between the IL and OL features where, lower value indicates better separation.

**Comparison with SOTA outlier detection methods:** The ability to accurately distinguish between IL and OL classes is key for improving the efficiency of ENVISE. We benchmark ENVISE on the CIFAR-100 and TI datasets with SOTA OL detection methods like error detection (ED) (Hendrycks & Gimpel (2016)), confidence estimation (CE) (DeVries & Taylor (2018)), confidence scaling (CS) (DeVries & Taylor (2018)), ODIN (Liang et al. (2018)), outlier exposure with confidence control (OECC) (Papadopoulos et al. (2019)), maximum classifier discrepancy (MCD) (Yu & Aizawa (2019)) and confidence aware learning (CAL) (Moon et al. (2020)). We use the official code of these methods and adaptively train them using their proposed loss functions on our experimental settings. For fair comparisons in terms of network architecture, we use the TN as DenseNet-201 and the SN as our binary VGG-16. Following Hendrycks & Gimpel (2016), we use FPR at 95% True positive rate (TPR), detection error (DE), area under ROC curve (AuROC), and area under precision-recall curve (AuPR) as our evaluation metric. From Table 2, we observe that ENVISE outperforms the best performing baseline method by achieving the lowest FPR and detection error with high AuROC and AuPR. SOTA model compression techniques (Frankle & Carbin (2019)) do not focus on processing images from the input stream with varying prior class probabilities. Hence, direct comparison with these methods is not meaningful since their objectives are different from those of ENVISE.

We visualize the separation of IL and OL classes in the feature space using UMAP (McInnes et al. (2018)) across the last fully connected layer of the BSN. Figure 4 shows that ENVISE has the best separation between IL and OL classes which is consistent with the quantitative analysis shown in Table 2. To quantify the feature separation, we compute the $p-$value using Wilcoxon's rank sum test (Wilcoxon (1992)) for the null hypothesis that the IL and OL feature distribution are the same i.e they overlap. Ideally for high separation, the *p-value* should be less than 0.05 (rejecting the hypothesis with 95% confidence). We observe from Figure 4 that the *p-value* of ENVISE has the smallest value as compared to the SOTA OL detection methods. This indicates that ENVISE achieves the least overlap between IL and OL classes, thereby outperforming the SOTA OL detection methods in its ability to better detect OL classes. A detailed comparison of ENVISE with each baseline method on different pairs of IL and OL classes and the corresponding feature space visualization is presented in the Appendix A.4. Furthermore, in Appendix A.2, we ablate the effect of the proposed $L_{at}$, margin ($\delta$) with the BSN

and also performance of OL detection with RvSN. We present additional discussions in Appendix A.3.

Table 2: Performance comparison of ENVISE with the best performing baseline method on different IL and OL class pairs on CIFAR-100 and TI dataset. ↓ and ↑ indicate smaller and greater value is better respectively. The detailed comparison with each baseline is in the Appendix A.4

| Dataset | Group | Method \ Metric | FPR (95% TPR) ↓ | Detection error ↓ | AuROC ↑ | AuPR (IL) ↑ | AuPR (OL) ↑ |
|---|---|---|---|---|---|---|---|
| CIFAR-100 | $C_1$ | Best Baseline. | 0.90 | 0.42 | 0.59 | 0.77 | 0.37 |
| | | ENVISE | **0.88** | **0.39** | **0.62** | **0.80** | **0.39** |
| | $C_2$ | Best Baseline | 0.69 | 0.28 | 0.79 | 0.86 | 0.71 |
| | | ENVISE | **0.41** | **0.16** | **0.89** | **0.88** | **0.92** |
| | $C_3$ | Best Baseline | 0.69 | 0.21 | 0.86 | 0.87 | 0.87 |
| | | ENVISE | **0.57** | **0.18** | **0.89** | **0.91** | **0.87** |
| | $C_4$ | Best Baseline | 0.65 | 0.27 | **0.79** | **0.85** | 0.72 |
| | | ENVISE | **0.61** | **0.26** | 0.75 | 0.82 | **0.77** |
| | $C_5$ | Best Baseline | 0.82 | 0.39 | 0.62 | 0.90 | 0.23 |
| | | ENVISE | **0.76** | **0.29** | **0.79** | **0.95** | **0.42** |
| | $C_6$ | Best Baseline | 0.63 | 0.22 | 0.85 | 0.79 | 0.90 |
| | | ENVISE | **0.49** | **0.16** | **0.91** | **0.87** | **0.97** |
| Tiny-Imagenet | $T_1$ | Best Baseline. | 0.72 | 0.28 | 0.84 | 0.88 | 0.65 |
| | | ENVISE | **0.69** | **0.23** | **0.89** | **0.91** | **0.71** |
| | $T_2$ | Best Baseline | 0.80 | 0.29 | 0.75 | 0.79 | 0.68 |
| | | ENVISE | **0.78** | **0.26** | **0.81** | **0.86** | **0.73** |
| | $T_3$ | Best Baseline | 0.78 | 0.32 | 0.68 | 0.76 | 0.61 |
| | | ENVISE | **0.74** | **0.20** | **0.81** | **0.93** | **0.63** |
| | $T_4$ | Best Baseline | 0.83 | 0.20 | 0.69 | 0.78 | **0.51** |
| | | ENVISE | **0.78** | **0.18** | **0.75** | **0.91** | 0.44 |
| | $T_5$ | Best Baseline | 0.80 | 0.30 | 0.90 | **0.85** | 0.79 |
| | | ENVISE | **0.78** | **0.26** | **0.91** | 0.85. | **0.84** |

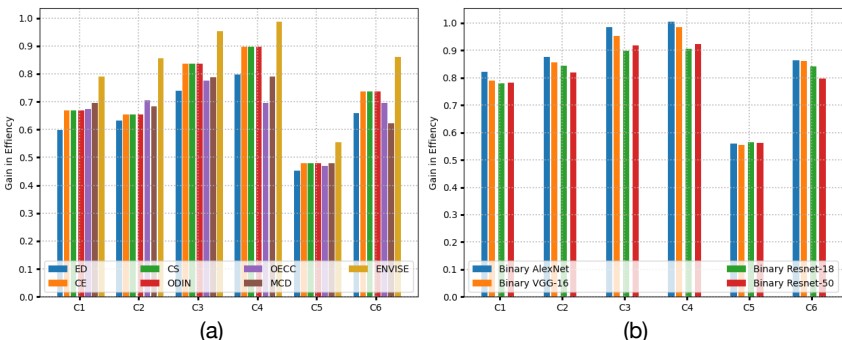

Figure 5: (a) Comparison of ENVISE with SOTA OL detection methods, and (b) Comparison of different BSN architectures in terms of GiE on different super-classes of the CIFAR-100 dataset.

**Gain in Efficiency (GiE):** The main focus of our work is to develop an efficient system to classify the image stream with low computational cost and high accuracy. We propose a new evaluation metric - GiE to measure the overall gain in efficiency from the number of times the BSN and TN are used individually to classify the image stream. During inference, we require the BSN to accurately classify the IL classes such that it does not rely on the TN thereby reducing the overall computation cost. Furthermore, the BSN should detect an image as OL so that the TN can be used in such cases. To process a single image on the CIFAR-100 dataset, the TN uses 9048 MFLOPs and 28.5msec, while the RvSN uses 310 MFLOPs and 3.04 msec. However, the BSN achieves 6 MFLOPs and 0.57 msec; a $\sim 50\times$ reduction in computation and a $\sim 6\times$ speed improvement over the RvSN. Furthermore, the BSN occupies $\sim 30\times$ and $\sim 100\times$ less memory than the RvSN and TN respectively.

During adaptive training, since each image is processed by both the BSN and TN, the total FLOPs used is $FL_{tr} = (X + Y)$, where $X$ and $Y$ are the FLOPs used by the TN and BSN respectively. Once the the BSN converges to the performance of the TN, only the BSN is used to process the input image. Hence, for IL images the total FLOPs is $FL_{in} = Y + (fp \times X)$, where $(fp \times X)$ indicates the number of times the TN is used when the BSN misclassifies an IL class as an OL. Similarly for OL images, the FLOPs is $FL_o = Y + (od \times X)$, where $(od \times X)$ denotes the OL classes correctly detected by the BSN which is then processed by the TN. Since the adaptive training and inference phase for a given super-class occur for $N_{tr}$ and $N_{in}$ epochs respectively, the *relative* FLOPS of

ENVISE compared to the TN is given as

$$RFL = \left[ \frac{N_{tr}}{N_T} \times \frac{FL_{tr}}{X} + \frac{N_{in}}{N_T} \times \frac{(p \times FL_{in} + q \times FL_o)}{X} \right] \tag{7}$$

Here, $N_T = N_{tr} + N_{in}$. For high computational efficiency, $RFL$ should be small. If the BSN works with perfect accuracy (i.e. $fp = 0$ and $od = 1.0$), then the minimum value of $RFL$ during inference is $RFL_{min} = \frac{Y + q \times X}{X}$. We define GiE as $RFL$ normalized with respect to its minimum value, i.e.

$$GiE = \frac{RFL_{min}}{RFL} = \frac{Y + q \times X}{\left[ \frac{N_{tr}}{N_T} \times FL_{tr} + \frac{N_{in}}{N_T} \times (p \times FL_{in} + q \times FL_o) \right]} \tag{8}$$

To numerically compute GiE, we use $X = 9048$ (TN), $Y = 6$ (BSN), $N_{tr} = 10$ and $N_{in} = 30$, while $p$ and $q$ are calculated from Table 1. However, as discussed previously, $N_{in}$ can be a very large value since the inference phase can occur for a very long duration. Figure 5 illustrates that ENVISE outperforms SOTA OL detection methods in terms of GiE for different scenarios on CIFAR-100 dataset. Thus, the $L_{at}$ ensures that the BSN can accurately distinguish between representations of IL and OL classes which enables ENVISE to operate in an efficient manner.

## 5 CONCLUSION

We propose ENVISE, an online distillation framework which uses a BSN to adaptively learn relevant knowledge from a TN to quickly classify the frequent classes in the deployed scenario. In doing so, it automatically allocates processing resources (i.e. either the BSN or the TN) to reduce overall computation. To learn proper representations of the IL classes, we propose an attention triplet loss that enables the BSN to learn the semantically relevant regions of the image that are emphasized by the TN. This enables the BSN to accurately distinguish between representations of IL and OL classes which is key for maintaining overall accuracy. Our experiments show that the BSN i) quickly converges to the performance of the TN, ii) classifies IL classes more accurately than a RvSN and other variants of the BSN, and iii) distinguishes between IL and OL classes with lower FPR and detection error than other SOTA OL detection methods on CIFAR-100 and tiny-imagenet datasets. We introduce a new metric GiE to assess the overall gain in efficiency, and experimentally show that the attention triplet loss enables ENVISE to achieve higher GiE than SOTA OL detection methods. We show that ENVISE is agnostic to the specific TN and BSN and achieves high gains with different BSN and TN architectures.

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

## A    APPENDIX

### A.1    PROOF OF CONVERGENCE

In Section 3 of the main paper, we describe that, one of the motivations for using the BSN instead of the RvSN is the ability of the BSN to adapt to the performance of the TN faster than the RvSN.

**Lemma A.1.** *Let RvSN and BSN represent the real-valued student network and binary student network respectively with the same architecture. Let $R$ denote the rate of convergence in terms of accuracy when the student network is adaptively trained using the teacher network's predictions. Then, $R(BSN) > R(RvSN)$ for the same input image stream and number of iterations.*

**Proof** For the ease of understanding, we assume to have a binary classification problem with 2 classes i.e. $C = 2$, the data points are linearly separable, the weights are initialized to 0 i.e. $w(0) = 0$ and learning rate $\eta = 1$. The initialization value of weights and learning rate do not affect the proof. Let us further assume that we have a stream of input images $x(n)$ where $n = 1, 2, 3...N$, where N is the total number of samples. Since the weights of the BSN are derived from the RvSN (From Section 3.1 of the main paper), we prove this lemma for RvSN first and then extend it to the BSN. Since, the pretrained performance of the SN is worse than the TN (comparing starting point of blue line with the red line from Figure 3(a) of the main paper), we assume that the SN misclassifies the images from the input stream i.e. $w(n)x(n) < 0$ where $w(n)$ is the weight of the network. The misclassification generates an error through which the network's weights are updated and there exists an optimal weight value $w^*$ such that the network converges i.e. misclassification error $= 0$. The weight update rule using back propagation is given by:

$$w(n + 1) = w(n) + \eta x(n)$$
$$[\text{ Since } \eta = 1] \quad w(n + 1) = w(n) + x(n) \tag{9}$$

Expanding eq (9), we have:

$$
\begin{aligned}
w(1) &= w(0) + x(0) \\
w(2) &= w(1) + x(1) \\
\Rightarrow w(2) &= w(0) + x(0) + x(1) \\
[\text{ Since } w(0) = 0] \quad w(2) &= x(0) + x(1)
\end{aligned}
\tag{10}
$$

Similarly we have $w(3) = x(2) + x(1) + x(0)$ and this recursively continues untill $w(n + 1)$. Thus,

$$
w(n + 1) = x(1) + x(2) + x(3) + ....x(n) \tag{11}
$$

Ideally, when the network converges, we obtain an optimal weight vector, which is represented as $w^\star$. We define the margin $\alpha$ as the shortest distance of $w^\star$ to the datapoint $x \in x(n)$. The concept of margin is similar to that of support vectors. Hence, the margin $\alpha$ is denoted as

$$
\alpha = \operatorname*{argmin}_{x(n) \in C} w^\star x(n) \tag{12}
$$

We ignore $x(0)$ for ease of calculation. Pre-multiplying both the sides of eq. (11) by $w^\star$, we have

$$
\begin{aligned}
w^\star w &= w^\star x(1) + w^\star x(2) + ....w^\star x(n) \\
[\text{From eq. (12)}] \qquad w^\star w &\ge n\alpha
\end{aligned}
\tag{13}
$$

From Cauchy - Schwarz inequality, we have

$$
\begin{aligned}
\|w^\star\|^2 \|w(n+1)\|^2 &\ge (w^\star w(n+1))^2 \\
\|w^\star\|^2 \|w(n+1)\|^2 &\ge (n\alpha)^2 \\
\boxed{\|w(n+1)\|^2 \ge \frac{n^2\alpha^2}{\|w^\star\|^2}}
\end{aligned}
\tag{14}
$$

We highlight eq. (14) as we will be using this in the later stages of the proof. We re-write eq. (9) in a different way as $w(k + 1) = x(k) + w(k) \ \ \forall \ k \in [0, 1, 2, 3..n]$. Squaring both sides of this equation and expanding it, we have

$$
\begin{aligned}
\|w(k+1)\|^2 &= \|x(k) + w(k)\|^2 \\
\Rightarrow \|w(k+1)\|^2 &= \|x(k)\|^2 + \|w(k)\|^2 + 2x(k)w(k) \\
[\text{Since in misclassification } w(n)x(n) \le 0] \quad \|w(k+1)\|^2 &\le \|x(k)\|^2 + \|w(k)\|^2 \\
\Rightarrow \|w(k+1)\|^2 - \|w(k)\|^2 &\le \|x(k)\|^2
\end{aligned}
\tag{15}
$$

Iterating eq (15) for all values of $k \in [0, 1, 2, 3..n]$, we have

$$
[\text{Since when } k = 0, w(k) = 0] \quad \|w(k+1)\|^2 \le \sum_{k=0}^{n} \|x(k)\|^2 \tag{16}
$$

Since $\|x(k)\|^2 > 0$, we define $\beta = \operatorname{argmax}_{x(n) \in C} \|x(k)\|^2$, thus eq. (9) can be re-written as

$$
\boxed{\|w(n+1)\|^2 \le n\beta} \tag{17}
$$

From eq. (14) and (17), we have

$$
\frac{n^2\alpha^2}{\|w^*\|^2} \le \|w(n+1)\|^2 \le n\beta \tag{18}
$$

From eq. (18), we can say that for this inequality to be satisfied, there exists $n^\star$ which denotes the optimal number of samples for which the network converges i.e. we obtain $w^\star$ at $n = n^\star$. This is given as follows:

$$
\begin{aligned}
\frac{n^{\star^2}\alpha^2}{\|w^\star\|^2} &= n^\star\beta \\
\boxed{n^\star = \frac{\beta \|w^\star\|^2}{\alpha^2}}
\end{aligned}
\tag{19}
$$

Thus from eq. (19), we can say that the network achieves convergence after being trained on $\frac{\beta\|w^*\|^2}{\alpha^2}$ number of samples. In the case of the BSN, the binary weights $\hat{w}^* = \Delta^* D^*$, where $\Delta^* = \frac{1}{N}|\hat{w}^\star|_1$ and $D^* = sign(w^*)$, Let $n_b^\star$ be the number of samples required for the BSN to converge. Thus, eq. (19) for the BSN would be given as

$$n_b^\star = \frac{\beta\|\hat{w}^\star\|^2}{\alpha^2}$$

$$n_b^\star = \frac{\beta\|\Delta^* D^*\|^2}{\alpha^2}$$

$$n_b^\star = \frac{\beta\|w^* D^*\|^2}{N^2\alpha^2}$$

$$n_b^\star = \frac{\beta\|w^\star\|^2 D^{*T}D^*}{N^2\alpha^2}$$

$$[\text{From eq. (19) and Rastegari et al. (2016)}] \quad n_b^\star = \frac{n^\star N}{N^2}$$

$$\boxed{n_b^\star = \frac{n^\star}{N}}$$

(20)

From eq. (19) and (20), we see that $n_b^* < n^*$ i.e. the BSN takes fewer samples than the RvSN to converge to the TN's performance given the same network architecture. We also experimentally observe the higher rate of convergence of the BSN over the RvSN in Figure 3 of the main paper.

## A.2 ABLATION STUDY

All the ablation studies are performed on the CIFAR-100 dataset. We illustrate the effectiveness of the attention triplet loss ($L_{at}$), the value of margin in the ($L_{at}$, significance of the hard negative term in $L_{at}$ and the performance of the BSN over the RvSN for OL detection. The quantitative analysis is summarized in Table 3.

**Effect of attention triplet loss ($L_{at}$) :** To test the effectiveness of the attention triplet loss ($L_{at}$), we train the BSN without it. Specifically, we use the TN as DenseNet-201, the BSN as Binary VGG-16 and train the BSN using only the distillation loss $L_d$. Comparing column ID 1 and 2 in Table 3, we observe that training the BSN using $L_d + L_{at}$ improves the mean FPR by 35.4%, mean detection error by 41.6%, mean AuROC by 12.5%, mean AuPR of IL class by 7.4% and mean AuPR of OL class by 30.3%. From Figure 6, we observe that the IL (red) and OL features (purple) are barely separable when evaluated on a pre-trained BSN. When the BSN is adaptively trained using only $L_d$, the features begin to separate but still have some overlap. However training the BSN using $L_d + L_{at}$ significantly enhances the feature separation thereby improving OL detection. Furthermore, the *p-value* (noted in Figure 6 below each UMAP visualizations) of BSN when trained using $L_d + L_{at}$ has the smallest value of 0.041. This validates our assertion that $L_{at}$ learns the proper representations of the IL classes thereby improving its ability to differentiate between IL and OL classes.

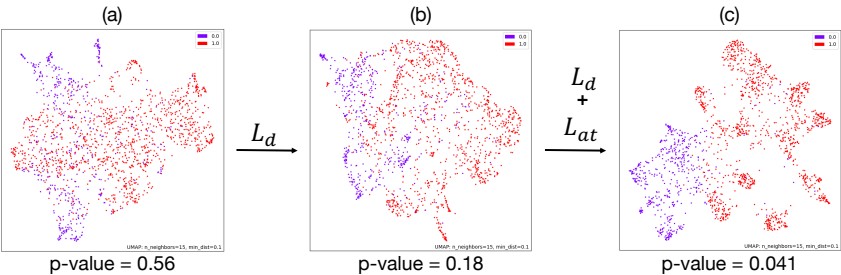

Figure 6: (a):(c) Illustration of the effect of applying $L_d$ and its combination with $L_{at}$ on the separation between the IL (red) and OL (purple) classes in the feature space.

**Effect of margin $\delta$ :** We mention in Section 3 that, in order to avoid a collapsed model early on during training, we select the hard negative attention map such that its squared distance is closest

to the hard positive attention map. Without using any OL images during training or validation, we empirically observe that $L_2$ norm of the hard negative attention map with respect to the hard positive attention map ranges between $[0.33, 2.8]$. Hence, we choose $\delta$ as 1.5 to ensure separability with respect to the hard positive attention map.

To show that ENVISE is insensitive to the specific value of margin, we train the BSN using $L_{at}$ with value of $\delta$ as 0.5, 1.0, 1.5, 2.0 and 2.5. From Table 3, comparing column 3 with 4,5,6,7 and 8, we observe that ENVISE is insensitive to the specific value of margin and outperforms the best performing baseline method for each super-class, even with different values of the margin. Furthermore, from Fig. 7, we observe that ENVISE has the best performance when the margin is set to 1.5

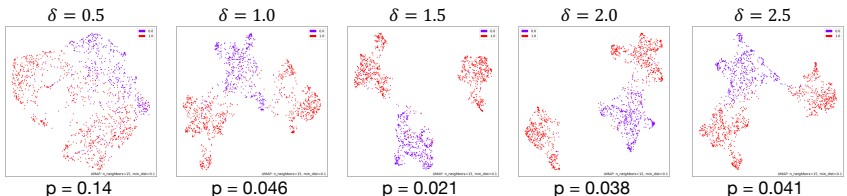

Figure 7: Effect of different values of margin on the separation between the IL (red) and OL (purple) classes in the feature space.

Table 3: Ablation study to illustrate the effect of proposed $L_{at}$ loss and the effect of different values of margin $\delta$ on the performance of ENVISE for outlier detection on CIFAR-100 dataset.

| Metric | super-class | ENVISE $L_d$ only | ENVISE $L_d + L_{at}$ | Best Baseline | ENVISE $\delta$=0.5 | ENVISE $\delta$=1.0 | ENVISE $\delta$=1.5 | ENVISE $\delta$=2.0 | ENVISE $\delta$=2.5 |
|---|---|---|---|---|---|---|---|---|---|
| column ID | | 1 | 2 | 3 | 4 | 5 | 6 | 7 | 8 |
| FPR (95% TPR) ↓ | $C_1$ | 0.93 | **0.88** | 0.90 | 0.95 | 0.89 | **0.88** | **0.88** | 0.93 |
| | $C_2$ | 0.81 | **0.41** | 0.69 | 0.49 | 0.61 | **0.41** | **0.41** | 0.41 |
| | $C_3$ | 0.9 | **0.57** | 0.69 | 0.58 | 0.66 | **0.57** | 0.65 | 0.83 |
| | $C_4$ | 0.88 | **0.61** | 0.65 | 0.84 | **0.61** | **0.61** | 0.79 | 0.78 |
| | $C_5$ | 0.87 | **0.76** | 0.82 | 0.78 | 0.81 | **0.76** | 0.83 | 0.77 |
| | $C_6$ | 0.64 | **0.49** | 0.63 | 0.54 | 0.53 | **0.49** | 0.67 | 0.57 |
| | mean | 0.84 | **0.62** | 0.73 | 0.70 | 0.69 | **0.62** | 0.69 | 0.72 |
| Detection error ↓ | $C_1$ | **0.39** | 0.40 | 0.42 | 0.46 | **0.33** | 0.40 | 0.38 | 0.41 |
| | $C_2$ | 0.33 | **0.16** | 0.28 | 0.17 | 0.21 | **0.16** | **0.16** | **0.16** |
| | $C_3$ | 0.35 | **0.18** | 0.21 | 0.22 | 0.25 | **0.18** | 0.2 | 0.24 |
| | $C_4$ | 0.35 | **0.26** | 0.27 | **0.26** | **0.26** | 0.3 | **0.26** | 0.3 |
| | $C_5$ | 0.39 | **0.29** | 0.39 | **0.29** | 0.33 | **0.29** | 0.31 | 0.3 |
| | $C_6$ | 0.21 | **0.16** | 0.22 | **0.16** | 0.17 | **0.16** | 0.25 | **0.16** |
| | mean | 0.34 | **0.24** | 0.30 | 0.27 | 0.26 | **0.24** | 0.26 | 0.26 |
| AuROC ↑ | $C_1$ | **0.64** | 0.62 | 0.59 | 0.54 | **0.71** | 0.62 | 0.66 | 0.6 |
| | $C_2$ | 0.83 | **0.89** | 0.79 | 0.9 | 0.87 | 0.89 | **0.95** | 0.91 |
| | $C_3$ | 0.69 | **0.89** | 0.86 | 0.84 | 0.83 | **0.89** | 0.87 | 0.82 |
| | $C_4$ | 0.67 | **0.75** | 0.79 | 0.76 | 0.78 | 0.75 | **0.80** | 0.76 |
| | $C_5$ | 0.64 | **0.79** | 0.62 | 0.76 | 0.72 | **0.79** | 0.74 | 0.76 |
| | $C_6$ | 0.87 | **0.91** | 0.85 | 0.89 | 0.89 | **0.91** | 0.83 | **0.91** |
| | mean | 0.72 | **0.81** | 0.75 | 0.78 | 0.80 | **0.81** | **0.81** | 0.79 |
| AuPR (inlier) ↑ | $C_1$ | **0.81** | 0.80 | 0.77 | 0.75 | **0.85** | 0.80 | 0.82 | 0.79 |
| | $C_2$ | 0.86 | **0.88** | 0.86 | 0.94 | 0.92 | 0.88 | 0.87 | **0.95** |
| | $C_3$ | 0.73 | **0.91** | 0.87 | 0.87 | 0.84 | **0.91** | 0.89 | 0.85 |
| | $C_4$ | 0.76 | **0.82** | 0.85 | 0.85 | **0.88** | 0.82 | 0.82 | 0.84 |
| | $C_5$ | 0.88 | **0.95** | 0.90 | 0.94 | 0.91 | **0.95** | 0.93 | 0.94 |
| | $C_6$ | 0.81 | **0.87** | 0.79 | 0.85 | 0.84 | **0.87** | 0.76 | 0.86 |
| | mean | 0.81 | **0.87** | 0.84 | **0.87** | **0.87** | **0.87** | 0.85 | 0.87 |
| AuPR (outlier) ↑ | $C_1$ | **0.41** | 0.39 | 0.37 | 0.32 | **0.45** | 0.39 | 0.43 | 0.35 |
| | $C_2$ | 0.65 | **0.92** | 0.71 | 0.82 | 0.76 | **0.92** | 0.88 | 0.84 |
| | $C_3$ | 0.64 | **0.93** | 0.87 | 0.78 | 0.81 | **0.93** | 0.85 | 0.77 |
| | $C_4$ | 0.52 | **0.77** | 0.72 | 0.61 | 0.62 | **0.77** | 0.68 | 0.63 |
| | $C_5$ | 0.26 | **0.42** | 0.23 | 0.38 | 0.33 | **0.42** | 0.34 | 0.38 |
| | $C_6$ | 0.88 | **0.97** | 0.90 | 0.92 | 0.92 | **0.97** | 0.88 | 0.94 |
| | mean | 0.56 | **0.73** | 0.59 | 0.63 | 0.65 | **0.73** | 0.68 | 0.65 |

Table 4: Ablation study to illustrate the effect of employing the BSN over RvSN on the performance of ENVISE for outlier detection on CIFAR-100 dataset.

| Method | mean FPR ($\downarrow$) | mean Detection error ($\downarrow$) | mean AuROC ($\uparrow$) | mean AuPR (IL) ($\uparrow$) | mean AuPR (OL) ($\uparrow$) |
|---|---|---|---|---|---|
| ENVISE (w/ RvSN) | 0.65 | 0.30 | 0.73 | 0.73 | 0.63 |
| ENVISE (w/ BSN) | **0.62** | **0.24** | **0.81** | **0.87** | **0.73** |

**Significance of using hard negative term in $L_{at}$:** As mentioned in Section 3 of the main paper, we observe that the attention maps from the correct and incorrect predictions are visually similar. This causes the BSN to learn improper representations of the IL classes during online distillation. Hence, we choose the attention map from the incorrect prediction (second most probable class) as a hard negative to ensure separability with the attention map from the correct prediction. To illustrate the significance of the hard negative attention map, we train ENVISE without the second term in equation 5 along with $L_d$. The attention $L_2$ loss is formulated as:

$$L_a = \frac{1}{N}\frac{1}{K}\sum_{n}^{N}(\sum_{k}^{K}\left\|A_{t_k} - A_{s_{p_k}}\right\|_2)$$ (21)

From Figure 8, we observe that the confidence of classifying an image decreases when ENVISE is trained using $L_a$. Furthermore, the attention map of ENVISE after training using $L_d + L_a$ does not focus on the semantically meaningful regions of the image (fourth column) unlike training it with $L_{at}$ (last column). Furthermore, we also observe training ENVISE with $L_d + L_a$ results in a poor OL detection with 0.98 FPR and 0.47 detection error respectively on $C_1$ as compared to 0.88 and 0.40 FPR and detection error respectively with $L_d + L_{at}$. This validates that the hard negative attention map ensures separability with the hard positive attention map and also improves the confidence of classification and OL detection.

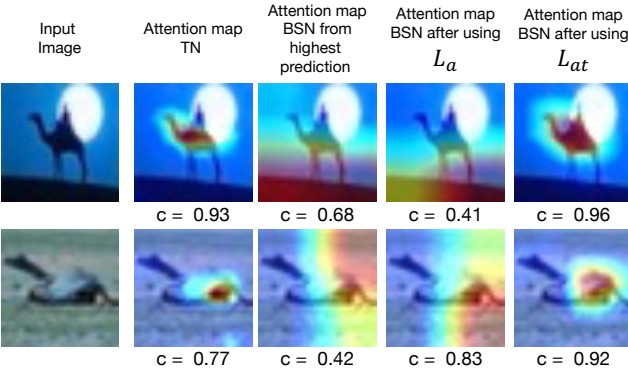

Figure 8: The attention map visualization representing the regions of the image that ENVISE focuses on when trained using $L_a$ and $L_{at}$. The confidence of classification is reported below each image.

**Effect of using RvSN instead of BSN** To illustrate the effectiveness of using the BSN to improve the overall gain in efficiency of the system, we employ a RvSN instead of the BSN as our SN. We use the TN as DenseNet-201 and real-valued VGG-16 as our SN and adaptively train it using $L_d + L_{at}$. We observe from Table 4 that using BSN instead of RvSN as our SN improves the mean FPR by 4.8%, mean detection error by 25%, mean AuROC by 11%, mean AuPR of IL class by 19.2% and mean AuPR of OL class by 15.9%. Furthermore, we also observe that mean GiE for the BSN is 0.53 while that of the RvSN is 0.78 which illustrates that ENVISE has 47.2% gain in efficiency with the BSN as compared to the RvSN.

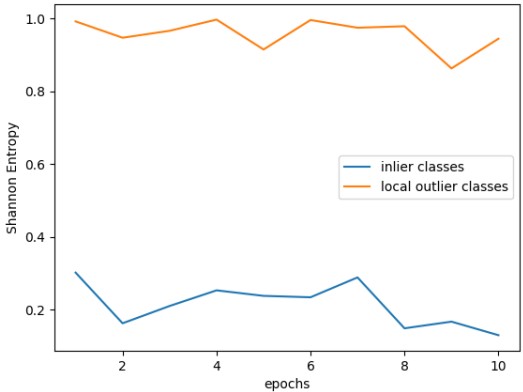

Figure 9: Shannon Entropy of IL and OL of the BSN during online distillation. The entropy of the IL decreases, the entropy of OL is high. This shows that BSN does not learn OL class information.

### A.3    DISCUSSIONS

**The TN does not leak OL class information to the BSN during online distillation** We mention in Section 3 that the BSN is trained using only the hard pseudo-labels from the TN on the IL classes. We show that during adaptive training, the TN does not transfer OL class information to the BSN. Hence, the ability of the BSN to detect OL classes is due to its ability to accurately differentiate between IL and OL class representations. To illustrate this, we compute the Shannon entropy $H(.)$ of OL and IL classes of the BSN during the online distillation process (when BSN is trained using $L_d + L_{at}$). Figure 9 illustrates that while the $H(\text{IL}|X)$ classes decreases, the $H(\text{OL}|X)$ class is already high, where $X$ is the image from the input stream. This illustrates that during the online distillation, the BSN learns the information of the IL classes, while information about the OL class is completely absent. Thus, the ability of the BSN to detect OL classes better than the SOTA outlier detection methods is not due to an OL information leak, but due to the BSN's ability to accurately differentiate between the representations of the IL and OL classes.

**Gain with smaller teacher network:** We investigate the importance of the architecture and size of the TN network on the performance of the BSN for efficient OL detection. Here, we use ResNet-18 (11M parameters) for the originally used DenseNet-201 (18M parameters) as the TN, which is a much smaller network as compared to the latter. We train ResNet-18 using the same training procedure as discussed in Section 4. Once the BSN converges to the performance of the TN, we evaluate the BSN on the different CIFAR-100 super-classes in Table 1 for OL detection. From Table 5, we observe that regardless of the TN used, ENVISE outperforms the best performing baseline method for OL detection by achieving lower FPR and detection error with higher AuROC and AuPR. Furthermore, we also observe that the BSN achieves similar gain in performance with ResNet-18 as with DenseNet-201. From Figure 3 and Table 5, we show that ENVISE is agnostic to the specific TN and BSN network architectures and achieves similar performance gains with different architectures.

Table 5: Performance of ENVISE in terms of OL detection, with a smaller TN network (ResNet-18) as compared larger TN (DenseNet-201) using the same BSN (Binary VGG-16) on different super-classes of CIFAR-100 dataset. Notations $R$ : ResNet-18 as TN, $D$ : DenseNet-201 as TN.

| Metric | Best Baseline $(R)$ | ENVISE $(R)$ | Best Baseline $(D)$ | ENVISE $(D)$ |
|---|---|---|---|---|
| mean FPR (95% TPR) ↓ | 0.83 | **0.69** | 0.76 | **0.62** |
| mean Detection error ↓ | 0.29 | **0.24** | 0.30 | **0.24** |
| mean AuROC ↑ | 0.74 | **0.80** | 0.74 | **0.81** |
| mean AuPR (IL) ↑ | 0.83 | **0.86** | 0.83 | **0.87** |
| mean AuPR (OL) ↑ | 0.61 | **0.81** | 0.63 | **0.73** |

**GiE when probability of IL class $>>$ probability of OL class:** We evaluate GiE of ENVISE for the case when the IL class occurs with a much higher probability than the OL class. Hence for $C_5$,

we consider all 2500 images from IL class and 300 images from OL class, making the probability of occurrence of IL class $p = 0.9$ ($\frac{2500}{2500+300} = 0.89 \sim 0.9$). In such a setting, we obtain an FPR ($f$) of 0.26 and detection error ($od$) of 0.15. From the formulation of $Fl_r$ mentioned in Section 4 of the main paper, we use $X = 9048 \times 1e^6$ and $Y = 6 \times 1e^6$ to obtain GiE = 0.79. Thus, in the case when the images from IL class occur more frequently than OL class (i.e $p = 0.9$), ENVISE achieves a gain in efficiency of 36.2%.

**Trade-off between accuracy and GiE in ENVISE:** Table 6 shows the accuracy of the BSN, the TN and their combined performance (i.e. ENVISE) for all six IL super-class pairs on the CIFAR-100 dataset. Comparing row 2 and row 3 in Table 6, we observe a very minimal loss in accuracy in ENVISE with respect to the TN on different super-classes of the CIFAR-100 dataset. In ENVISE, during inference, the BSN classifies the IL classes from the image stream. When the BSN misclassifies an IL class as an OL class, the image is then given to the TN for classification. This ensures that the overall accuracy of the system is bounded by the performance of the TN. Furthermore, from Figure 5, we observe that ENVISE has the highest GiE which indicates that, along with high computational efficiency, ENVISE also has minimal loss in overall accuracy as compared to a standalone TN.

Table 6: Comparison of accuracy between BSN, TN and ENVISE on the IL classes of CIFAR-100 dataset. The IL images that the BSN misclassifies as OD, are correctly reclassified by the TN.

| Model | $C_1$ | $C_2$ | $C_3$ | $C_4$ | $C_5$ | $C_6$ |
|---|---|---|---|---|---|---|
| BSN | 0.72 | 0.80 | 0.86 | 0.83 | 0.79 | 0.83 |
| TN | 0.76 | 0.81 | 0.89 | 0.86 | 0.83 | 0.84 |
| ENVISE | 0.75 | 0.81 | 0.88 | 0.84 | 0.81 | 0.84 |

### A.4 ADDITIONAL QUANTITATIVE AND QUALITATIVE ANALYSIS

**Complete comparison with baseline methods:** We present a detailed comparison of ENVISE with the SOTA OL methods on CIFAR-100 and TI datasets in Table 9. We observe that ENVISE achieves lower FPR and detection error as compared to the baseline methods. Specifically, ENVISE outperforms the best performing baseline method (ODIN) by 23% and 25% in terms of mean FPR and mean detection error respectively. Furthermore, ENVISE also achieves higher AuROC and AuPR (outlier class) by outperforming ODIN by 9.5% and 15.9% in terms of mean AuROC and mean AuPR (outlier class) respectively on CIFAR-100 dataset. Table 9 also shows that ENVISE outperforms the baseline methods achieving lower FPR and detection error and higher AuROC and AuPR (IL and OL class) on TI dataset. Specifically, ENVISE achieve 9.3% and 30.4% lower mean FPR and mean detection error respectively as compared to the best performing baseline method (MCD). ENVISE also outperforms MCD by 5.5%, 14.1% and 3.1% in terms of mean AuROC, mean AuPR (inlier class) and mean AuPR (outlier class) respectively.

**Performance of ENVISE is insensitive to specific IL \ OL class pairs:** As mentioned in Table 1, we evaluate ENVISE on meaningful IL \ OL class pairs that mimic real-life scenarios. However, we show that ENVISE is insensitive to the specific IL \ OL class pairs created in Table 1. From Table 10, Table 11 and Table 12, we show that ENVISE outperforms all baseline methods when each IL class is compared to every other OL class by achieving lower FPR and detection error and higher AuROC and AuPR on the CIFAR-100 dataset.

**Feature space separation:** Figure 10 illustrates the comparison of ENVISE with all baselines in terms of the separation between IL and OL images in the feature space for all IL \ OL class pairs on CIFAR-100 dataset. To quantify the feature separation between the IL and OL samples, we compute the $p-$value using Wilcoxon's rank sum test Wilcoxon (1992) for the null hypothesis that the IL and OL feature distribution are the same. Hence, lower $p-$value indicates better separation. We observe that ENVISE achieves better separation than baseline methods with the lowest $p-$value. Furthermore, we also observe clusters of the IL images (in red) in the feature space which illustrates that ENVISE is capable of learning representations of IL images rather than memorizing labels from the TN.

**Sub-classes within each super-class:** Table 7 and Table 8 represents the sub-classes within each super-class of the CIFAR-100 and tiny-imagenet (TI) datasets. These sub-classes are the original classes of these datasets respectively. Each super-class is denoted by $C_{xi}$ and $C_{xo}$ where $x = [1, 2, 3, 4, 5, 6]$ representing the inlier and outlier super-class on CIFAR-100 dataset. Furthermore, the

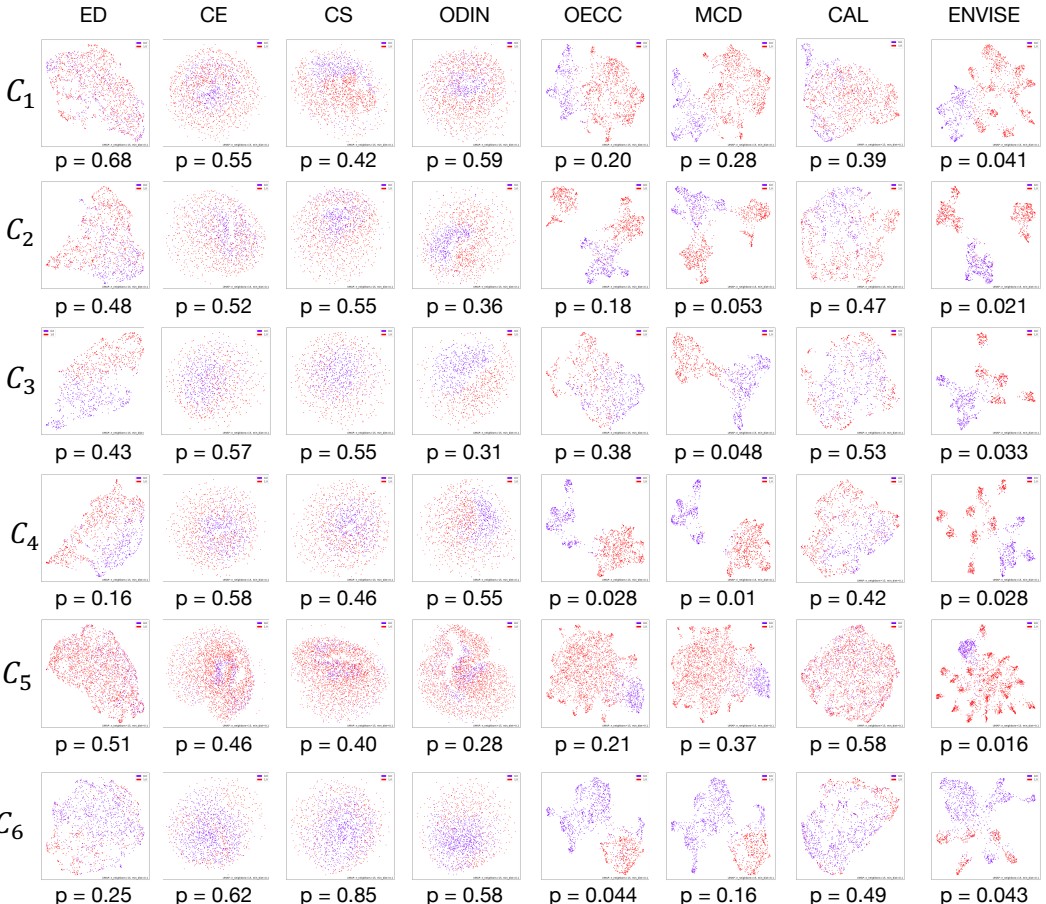

Figure 10: Comparison of ENVISE with baseline methods in terms of separation between IL (in red) and OL (in purple) samples in the feature space on CIFAR-100 dataset. The $p-$value obtained from Wilcoxon's rank sum test is denoted below each visualization where lower value denotes better separation.

super-classes of the tiny-imagenet dataset are denoted as $T_{yi}$ and $T_{yo}$ for inlier and outlier super-class, where $y = [1, 2, 3, 4, 5]$. We use these notations for comparing ENVISE with the SOTA OL detection methods in Table 9, Table 10, Table 11 and Table 12 respectively.

Table 7: Different sub-classes within each *inlier* and *outlier* super-classes on the CIFAR-100 dataset. These sub-classes represent the original classes in the CIFAR-100 dataset.

| Super-classes | Sub-classes |
|---|---|
| water animals ($C_{1i}$) | beaver, dolphin, otter, seal, whale, aquarium fish, flatfish, ray shark, trout, crab, lobster |
| food containers ($C_{1o}$) | bottles, bowls, caps, cups, plates |
| flora ($C_{2i}$) | orchids, poppies, roses, sunflower tulips, maple, oak, palm, pine, willow |
| household electric devices ($C_{2o}$) | clock, keyboard, lamp, telephone, television |
| fruit and vegetables ($C_{3i}$) | apple, mushrooms, orange, pears, sweet peppers |
| household furniture ($C_{3o}$) | bed, chair, couch, table, wardrobe |
| insects ($C_{4i}$) | bee, beetle, butterfly, caterpillar, cockroach, snail, spider, worm |
| manmade things ($C_{4o}$) | bridge, castle, house, road, skyscrapper |
| animals ($C_{5i}$) | camel, cattle, chimpanzee, elephant, kangaroo, fox, porcupine, possum racoon,skunk, hamster, mouse, rabbit, shrew, squirel, crocodile dinosaur, lizard, turtle, snake, bear, leopard, tiger, lion, wolf |
| people ($C_{5o}$) | baby, boy, girl, man, women |
| outdoor scene ($C_{6i}$) | cloud, forest, mountain, plain, sea |
| vehicles ($C_{6o}$) | bicycle, bus, motorcycle, pickup truck train, lawn-mover rocket, streetcar tank, tractor |

Table 8: Different sub-classes within each *inlier* and *outlier* super-classes on the tiny-imagenet dataset. These sub-classes represent the original classes in the tiny-imagenet dataset.

| Super-classes | Sub-classes |
|---|---|
| animals ($T_{1i}$) | Labrador retriever, hog, Chihuahua, orangutan, chimpanzee, koala, lesser panda golden retriever, lion, baboon, African elephant, Yorkshire terrier, bison, standard poodle, cougar, gazelle, ox, Egyptian cat Persian cat, German shepherd, guinea pig, bighorn |
| birds ($T_{1o}$) | black stork, goose, king penguin, albatross, crane |
| reptiles / insects ($T_{2i}$) | European fire salamander, bee, scorpion, fly, black widow, ladybug, slug, cockroach, grasshopper, centipede, mantis, sulphur butterfly dragonfly, snail, boa constrictor, spider web, trilobite, tarantula |
| aquatic animals ($T_{2o}$) | brain coral, American lobster, American alligator, tailed frog, spiny lobster sea slug, dugong, coral reef, bullfrog, sea cucumber, jellyfish, goldfish |
| edible items ($T_{3i}$) | orange, pizza, mushroom, banana, ice cream, pomegranate, pretzel, mashed potato potpie, cauliflower, meat loaf, bell pepper, guacamole, lemon, ice lolly, confectionery |
| garments ($T_{3o}$) | swimming trunks, apron, poncho, academic gown, military uniform, neck brace vestment, kimono, sombrero, fur coat, cardigan, bikini, miniskirt, bow tie sunglasses, sandal, sock, Christmas stocking |
| Household items ($T_{4i}$) | picket fence, candle, chain, rocking chair, torch, iPod frying pan, dumbbell, water jug, teddy, plate, walking stick, computer keyboard bucket, comic book, sewing machine, remote control, teapot, barn, volleyball lawn mower, hourglass, desk, lampshade, bathtub, wok, CD player, rugby ball stopwatch, magnetic compass, space heater, plunger, backpack, wooden spoon broom, dining table, basketball, punching bag, refrigerator, umbrella |
| places / manmade things ($T_{4o}$) | seashore, water tower, triumphal arch, cliff dwelling, suspension bridge dam, steel arch bridge, cliff, monarch, obelisk, lakeside, fountain, altar, alp |
| Vehicles ($T_{5i}$) | school bus, lifeboat, jinrikisha, beach wagon, tractor, moving van, trolleybus police van, go kart, freight car, bullet train, sports car, gondola, limousine, convertible |
| Miscellaneous ($T_{5o}$) | beaker, parking meter, drumstick, reel, potter's wheel, beacon, acorn, gasmask, projectile cannon, cash machine, maypole, pay phone, flagpole, bannister, thatch, pill, bottle, pop bottle barbershop, birdhouse, binoculars, organ, abacus, nail turnstile, beer bottle, oboe viaduct, scoreboard, barrel, pole, syringe, chest, butcher shop, espresso, snorkel, brass |

Table 9: Performance comparison of ENVISE with the baseline methods on different IL and OL classes on CIFAR-100 and TI datasets as presented in Table 2. The ↓ indicates smaller value is better and ↑ indicates greater value is better.

| Metric | inlier \ outlier class | ED | CE | CS | ODIN | OECC | MCD | CAL | ENVISE | inlier \ outlier class | ED | CE | CS | ODIN | OECC | MCD | CAL | ENVISE |
|---|---|---|---|---|---|---|---|---|---|---|---|---|---|---|---|---|---|---|
| FPR (95% TPR) ↓ | $C_{1i}\backslash C_{1o}$ | 0.93 | 0.90 | 0.90 | 0.90 | 0.90 | 0.95 | 0.95 | **0.88** | $T_{1i}\backslash T_{1o}$ | 0.72 | 0.99 | 0.98 | 0.98 | 0.81 | 0.87 | 0.93 | **0.69** |
| | $C_{2i}\backslash C_{2o}$ | 0.91 | 0.89 | 0.72 | 0.72 | 0.86 | 0.85 | 0.69 | **0.41** | $T_{2i}\backslash T_{2o}$ | 0.87 | 0.98 | 1.0 | 0.87 | 0.84 | 0.80 | 0.87 | **0.78** |
| | $C_{3i}\backslash C_{3o}$ | 0.83 | 0.70 | 0.70 | 0.72 | 0.90 | 0.89 | 0.69 | **0.57** | $T_{3i}\backslash T_{3o}$ | 0.78 | 0.89 | 0.82 | 0.82 | 0.82 | 0.81 | 0.85 | **0.74** |
| | $C_{4i}\backslash C_{4o}$ | 0.91 | 0.65 | 0.66 | 0.66 | 0.97 | 0.94 | 0.87 | **0.61** | $T_{4i}\backslash T_{4o}$ | 0.94 | 1.0 | 1.0 | 0.83 | 0.87 | 0.84 | 0.92 | **0.78** |
| | $C_{5i}\backslash C_{5o}$ | 0.94 | 0.92 | 0.93 | 0.92 | 0.90 | 0.95 | 0.82 | **0.76** | $T_{5i}\backslash T_{5o}$ | 0.88 | 0.88 | 0.87 | 0.82 | 0.90 | 0.80 | 0.83 | **0.78** |
| | $C_{6i}\backslash C_{6o}$ | 0.83 | 0.69 | 0.68 | 0.66 | 0.89 | 0.95 | 0.63 | **0.49** | | | | | | | | | |
| | mean | 0.89 | 0.79 | 0.76 | 0.76 | 0.90 | 0.92 | 0.77 | **0.62** | - | 0.84 | 0.95 | 0.93 | 0.86 | 0.85 | 0.82 | 0.88 | **0.75** |
| Detection error ↓ | $C_{1i}\backslash C_{1o}$ | 0.42 | 0.46 | 0.46 | 0.45 | 0.43 | 0.45 | 0.45 | **0.39** | $T_{1i}\backslash T_{1o}$ | 0.39 | 0.48 | 0.47 | 0.43 | 0.28 | 0.38 | 0.39 | **0.23** |
| | $C_{2i}\backslash C_{2o}$ | 0.33 | 0.28 | 0.29 | 0.28 | 0.39 | 0.35 | 0.41 | **0.16** | $T_{2i}\backslash T_{2o}$ | 0.40 | 0.43 | 0.42 | 0.31 | 0.29 | 0.31 | 0.34 | **0.26** |
| | $C_{3i}\backslash C_{3o}$ | 0.32 | 0.22 | 0.21 | 0.22 | 0.50 | 0.50 | 0.39 | **0.18** | $T_{3i}\backslash T_{3o}$ | 0.45 | 0.39 | 0.41 | 0.32 | 0.37 | 0.32 | 0.43 | **0.20** |
| | $C_{4i}\backslash C_{4o}$ | 0.42 | 0.27 | 0.28 | 0.27 | 0.45 | 0.42 | 0.42 | **0.26** | $T_{4i}\backslash T_{4o}$ | 0.29 | 0.31 | 0.33 | 0.38 | 0.51 | 0.20 | 0.32 | **0.18** |
| | $C_{5i}\backslash C_{5o}$ | 0.41 | 0.39 | 0.39 | 0.39 | 0.46 | 0.44 | 0.44 | **0.29** | $T_{5i}\backslash T_{5o}$ | 0.34 | 0.37 | 0.34 | 0.34 | 0.36 | 0.30 | 0.36 | **0.26** |
| | $C_{6i}\backslash C_{6o}$ | 0.29 | 0.22 | 0.22 | 0.22 | 0.43 | 0.40 | 0.42 | **0.16** | | | | | | | | | |
| | mean | 0.37 | 0.31 | 0.31 | 0.30 | 0.44 | 0.43 | 0.42 | **0.24** | - | 0.37 | 0.40 | 0.39 | 0.36 | 0.36 | 0.30 | 0.37 | **0.23** |
| AuROC ↑ | $C_{1i}\backslash C_{1o}$ | 0.56 | 0.54 | 0.54 | 0.54 | 0.59 | 0.54 | 0.55 | **0.62** | $T_{1i}\backslash T_{1o}$ | 0.72 | 0.56 | 0.55 | 0.58 | 0.55 | 0.84 | 0.79 | **0.89** |
| | $C_{2i}\backslash C_{2o}$ | 0.71 | 0.78 | 0.77 | 0.79 | 0.62 | 0.68 | 0.59 | **0.89** | $T_{2i}\backslash T_{2o}$ | 0.75 | 0.55 | 0.55 | 0.65 | 0.66 | 0.71 | 0.72 | **0.81** |
| | $C_{3i}\backslash C_{3o}$ | 0.73 | 0.85 | 0.86 | 0.85 | 0.45 | 0.43 | 0.60 | **0.89** | $T_{3i}\backslash T_{3o}$ | 0.65 | 0.51 | 0.50 | 0.58 | 0.64 | 0.68 | 0.65 | **0.81** |
| | $C_{4i}\backslash C_{4o}$ | 0.59 | 0.78 | **0.85** | **0.79** | 0.53 | 0.57 | 0.59 | 0.75 | $T_{4i}\backslash T_{4o}$ | 0.69 | 0.54 | 0.53 | 0.64 | 0.64 | 0.69 | 0.69 | 0.75 |
| | $C_{5i}\backslash C_{5o}$ | 0.59 | 0.62 | 0.62 | 0.62 | 0.50 | 0.56 | 0.54 | **0.79** | $T_{5i}\backslash T_{5o}$ | 0.90 | 0.55 | 0.54 | 0.69 | 0.61 | 0.73 | 0.90 | **0.91** |
| | $C_{6i}\backslash C_{6o}$ | 0.78 | 0.85 | 0.84 | 0.85 | 0.57 | 0.60 | 0.60 | **0.91** | | | | | | | | | |
| | mean | 0.66 | 0.74 | 0.74 | 0.74 | 0.54 | 0.56 | 0.58 | **0.81** | - | 0.74 | 0.54 | 0.53 | 0.63 | 0.62 | 0.73 | 0.75 | **0.77** |
| AuPR (inlier) ↑ | $C_{1i}\backslash C_{1o}$ | 0.73 | 0.74 | 0.74 | 0.74 | 0.77 | 0.74 | 0.75 | **0.80** | $T_{1i}\backslash T_{1o}$ | 0.78 | 0.61 | 0.61 | 0.66 | 0.63 | 0.88 | 0.86 | **0.91** |
| | $C_{2i}\backslash C_{2o}$ | 0.83 | 0.86 | 0.85 | 0.86 | 0.73 | 0.79 | 0.71 | **0.88** | $T_{2i}\backslash T_{2o}$ | 0.79 | 0.56 | 0.56 | 0.66 | 0.58 | 0.66 | 0.78 | **0.86** |
| | $C_{3i}\backslash C_{3o}$ | 0.75 | 0.85 | 0.87 | 0.85 | 0.47 | 0.45 | 0.63 | **0.91** | $T_{3i}\backslash T_{3o}$ | 0.63 | 0.71 | 0.71 | 0.69 | 0.75 | 0.76 | 0.76 | **0.93** |
| | $C_{4i}\backslash C_{4o}$ | 0.71 | 0.84 | **0.85** | 0.84 | 0.66 | 0.65 | 0.67 | 0.82 | $T_{4i}\backslash T_{4o}$ | 0.68 | 0.75 | 0.75 | 0.74 | 0.77 | 0.78 | 0.77 | **0.91** |
| | $C_{5i}\backslash C_{5o}$ | 0.87 | 0.90 | 0.90 | 0.90 | 0.85 | 0.87 | 0.85 | **0.95** | $T_{5i}\backslash T_{5o}$ | 0.85 | 0.32 | 0.31 | 0.87 | 0.69 | 0.84 | **0.85** | **0.85** |
| | $C_{6i}\backslash C_{6o}$ | 0.66 | 0.79 | 0.79 | 0.79 | 0.39 | 0.43 | 0.44 | **0.87** | | | | | | | | | |
| | mean | 0.76 | 0.83 | 0.83 | 0.83 | 0.65 | 0.66 | 0.67 | **0.87** | - | 0.74 | 0.59 | 0.59 | 0.72 | 0.68 | 0.78 | 0.80 | **0.89** |
| AuPR (outlier) ↑ | $C_{1i}\backslash C_{1o}$ | 0.35 | 0.35 | 0.35 | 0.36 | 0.37 | 0.31 | 0.32 | **0.39** | $T_{1i}\backslash T_{1o}$ | 0.38 | 0.48 | 0.47 | 0.48 | 0.65 | 0.65 | 0.65 | **0.71** |
| | $C_{2i}\backslash C_{2o}$ | 0.55 | 0.70 | 0.70 | 0.71 | 0.48 | 0.52 | 0.44 | **0.92** | $T_{2i}\backslash T_{2o}$ | 0.50 | 0.52 | 0.51 | 0.64 | 0.59 | 0.68 | 0.67 | **0.73** |
| | $C_{3i}\backslash C_{3o}$ | 0.73 | 0.86 | 0.87 | 0.86 | 0.46 | 0.45 | 0.56 | **0.93** | $T_{3i}\backslash T_{3o}$ | 0.47 | 0.29 | 0.29 | 0.47 | 0.44 | 0.61 | 0.59 | **0.63** |
| | $C_{4i}\backslash C_{4o}$ | 0.46 | 0.72 | 0.70 | 0.72 | 0.40 | 0.46 | 0.49 | **0.77** | $T_{4i}\backslash T_{4o}$ | 0.29 | 0.29 | 0.28 | 0.45 | 0.43 | **0.51** | **0.51** | 0.44 |
| | $C_{5i}\backslash C_{5o}$ | 0.23 | 0.21 | 0.21 | 0.21 | 0.16 | 0.19 | 0.19 | **0.42** | $T_{5i}\backslash T_{5o}$ | 0.79 | 0.73 | 0.74 | 0.79 | 0.57 | 0.78 | 0.78 | **0.84** |
| | $C_{6i}\backslash C_{6o}$ | 0.86 | 0.90 | 0.89 | 0.90 | 0.70 | 0.74 | 0.74 | **0.97** | | | | | | | | | |
| | mean | 0.53 | 0.62 | 0.62 | 0.63 | 0.43 | 0.44 | 0.46 | **0.73** | - | 0.49 | 0.46 | 0.46 | 0.57 | 0.54 | 0.65 | 0.64 | **0.67** |

Table 10: Comparison of ENVISE with the baseline methods on IL class pairs $C_1$ and $C_2$ with every OL class on CIFAR-100. The ↓ indicates smaller value is better and ↑ indicates greater value is better.

| Metric | inlier \ outlier class | ED | CE | CS | ODIN | OECC | MCD | CAL | ENVISE | inlier \ outlier class | ED | CE | CS | ODIN | OECC | MCD | CAL | ENVISE |
|---|---|---|---|---|---|---|---|---|---|---|---|---|---|---|---|---|---|---|
| FPR (95% TPR) ↓ | $C_{1i} \backslash C_{1o}$ | 0.93 | 0.90 | 0.90 | 0.90 | 0.90 | 0.95 | 0.95 | **0.88** | $C_{2i} \backslash C_{1o}$ | 0.92 | 0.82 | 0.89 | 0.87 | 0.79 | 0.86 | 0.85 | **0.69** |
| | $C_{1i} \backslash C_{2o}$ | 0.95 | 0.97 | 0.96 | 0.81 | 0.90 | 0.97 | 0.86 | **0.68** | $C_{2i} \backslash C_{2o}$ | 0.91 | 0.89 | 0.72 | 0.72 | 0.86 | 0.85 | 0.69 | **0.41** |
| | $C_{1i} \backslash C_{3o}$ | 0.94 | 1.0 | 1.0 | 0.81 | 0.93 | 0.98 | 0.76 | **0.68** | $C_{2i} \backslash C_{3o}$ | 0.93 | 0.89 | 0.88 | 0.89 | 0.86 | 0.89 | 0.81 | **0.70** |
| | $C_{1i} \backslash C_{4o}$ | 0.94 | 0.99 | 0.84 | 0.83 | 0.94 | 0.89 | 0.83 | **0.76** | $C_{2i} \backslash C_{4o}$ | 0.95 | 0.92 | 0.93 | 0.73 | 0.87 | 0.86 | 0.78 | **0.68** |
| | $C_{1i} \backslash C_{5o}$ | 0.94 | 0.99 | 0.89 | 0.78 | 0.83 | 0.88 | 0.77 | **0.42** | $C_{2i} \backslash C_{5o}$ | 0.92 | 0.89 | 0.88 | 0.96 | 0.73 | 0.86 | 0.76 | **0.45** |
| | $C_{1i} \backslash C_{6o}$ | 0.94 | 0.80 | 0.90 | 0.71 | 0.92 | 0.81 | 0.83 | **0.79** | $C_{2i} \backslash C_{6o}$ | 0.92 | 0.86 | 0.86 | 0.83 | 0.83 | 0.87 | 0.88 | **0.69** |
| | mean | 0.94 | 0.94 | 0.92 | 0.81 | 0.91 | 0.91 | 0.83 | **0.70** | - | 0.92 | 0.85 | 0.86 | 0.83 | 0.82 | 0.87 | 0.83 | **0.60** |
| Detection error ↓ | $C_{1i} \backslash C_{1o}$ | 0.42 | 0.46 | 0.46 | 0.45 | 0.43 | 0.45 | 0.45 | **0.39** | $C_{2i} \backslash C_{1o}$ | 0.41 | 0.41 | 0.41 | 0.33 | 0.33 | 0.38 | 0.41 | **0.31** |
| | $C_{1i} \backslash C_{2o}$ | 0.50 | 0.49 | 0.49 | 0.35 | 0.40 | 0.40 | 0.38 | **0.31** | $C_{2i} \backslash C_{2o}$ | 0.33 | 0.28 | 0.29 | 0.28 | 0.39 | 0.35 | 0.41 | **0.16** |
| | $C_{1i} \backslash C_{3o}$ | 0.49 | 0.50 | 0.50 | 0.38 | 0.44 | 0.48 | 0.36 | **0.32** | $C_{2i} \backslash C_{3o}$ | 0.44 | 0.45 | 0.45 | 0.41 | 0.40 | 0.38 | 0.37 | **0.36** |
| | $C_{1i} \backslash C_{4o}$ | 0.46 | 0.50 | 0.42 | 0.41 | 0.45 | 0.37 | 0.35 | **0.32** | $C_{2i} \backslash C_{4o}$ | 0.47 | 0.45 | 0.45 | 0.39 | 0.39 | 0.43 | 0.38 | **0.34** |
| | $C_{1i} \backslash C_{5o}$ | 0.49 | 0.50 | 0.50 | 0.36 | 0.27 | 0.34 | 0.29 | **0.18** | $C_{2i} \backslash C_{5o}$ | 0.39 | 0.43 | 0.45 | 0.32 | 0.24 | 0.31 | 0.28 | **0.21** |
| | $C_{1i} \backslash C_{6o}$ | 0.49 | 0.43 | 0.40 | 0.35 | 0.41 | 0.37 | 0.38 | **0.35** | $C_{2i} \backslash C_{6o}$ | 0.43 | 0.47 | 0.46 | **0.34** | 0.36 | 0.40 | 0.39 | 0.35 |
| | mean | 0.48 | 0.48 | 0.46 | 0.39 | 0.40 | 0.40 | 0.37 | **0.31** | - | 0.42 | 0.42 | 0.43 | 0.35 | 0.35 | 0.38 | 0.37 | **0.23** |
| AuROC → | $C_{1i} \backslash C_{1o}$ | 0.56 | 0.54 | 0.54 | 0.54 | 0.59 | 0.54 | 0.55 | **0.62** | $C_{2i} \backslash C_{1o}$ | 0.61 | 0.53 | 0.57 | 0.68 | 0.72 | 0.69 | 0.68 | **0.74** |
| | $C_{1i} \backslash C_{2o}$ | 0.46 | 0.47 | 0.49 | 0.74 | 0.62 | 0.64 | 0.71 | **0.76** | $C_{2i} \backslash C_{2o}$ | 0.71 | 0.78 | 0.77 | 0.79 | 0.62 | 0.68 | 0.59 | **0.89** |
| | $C_{1i} \backslash C_{3o}$ | 0.41 | 0.31 | 0.39 | 0.63 | 0.56 | **0.68** | 0.67 | 0.64 | $C_{2i} \backslash C_{3o}$ | 0.54 | 0.60 | 0.59 | 0.61 | 0.62 | 0.60 | 0.71 | **0.74** |
| | $C_{1i} \backslash C_{4o}$ | 0.52 | 0.50 | 0.45 | 0.71 | 0.55 | 0.63 | 0.70 | **0.79** | $C_{2i} \backslash C_{4o}$ | 0.54 | 0.53 | 0.60 | 0.59 | **0.64** | 0.57 | 0.61 | 0.62 |
| | $C_{1i} \backslash C_{5o}$ | 0.47 | 0.23 | 0.64 | 0.71 | 0.77 | 0.79 | 0.83 | **0.89** | $C_{2i} \backslash C_{5o}$ | 0.61 | 0.64 | 0.64 | 0.77 | 0.83 | 0.77 | 0.76 | **0.85** |
| | $C_{1i} \backslash C_{6o}$ | 0.49 | 0.67 | 0.67 | 0.63 | 0.58 | 0.69 | 0.69 | **0.73** | $C_{2i} \backslash C_{6o}$ | 0.58 | 0.51 | 0.50 | 0.62 | 0.68 | 0.64 | 0.63 | **0.70** |
| | mean | 0.48 | 0.45 | 0.53 | 0.66 | 0.61 | 0.67 | 0.70 | **0.74** | - | 0.58 | 0.59 | 0.61 | 0.68 | 0.69 | 0.66 | 0.68 | **0.71** |
| AuPR (inlier) → | $C_{1i} \backslash C_{1o}$ | 0.73 | 0.74 | 0.74 | 0.74 | 0.77 | 0.74 | 0.75 | **0.80** | $C_{2i} \backslash C_{1o}$ | 0.64 | 0.63 | 0.63 | 0.61 | 0.61 | 0.63 | 0.69 | **0.77** |
| | $C_{1i} \backslash C_{2o}$ | 0.67 | 0.68 | 0.69 | 0.81 | 0.77 | 0.70 | 0.75 | **0.84** | $C_{2i} \backslash C_{2o}$ | 0.83 | 0.86 | 0.85 | 0.86 | 0.73 | 0.79 | 0.71 | **0.88** |
| | $C_{1i} \backslash C_{3o}$ | 0.64 | 0.60 | 0.63 | 0.71 | 0.71 | 0.57 | 0.68 | **0.73** | $C_{2i} \backslash C_{3o}$ | 0.61 | 0.60 | 0.60 | 0.71 | 0.71 | 0.67 | 0.69 | **0.73** |
| | $C_{1i} \backslash C_{4o}$ | **0.74** | 0.68 | 0.68 | 0.78 | 0.73 | 0.65 | 0.74 | 0.71 | $C_{2i} \backslash C_{4o}$ | 0.62 | 0.57 | 0.57 | 0.63 | 0.61 | 0.66 | 0.58 | **0.67** |
| | $C_{1i} \backslash C_{5o}$ | 0.68 | 0.58 | 0.58 | 0.76 | 0.88 | 0.82 | 0.83 | **0.94** | $C_{2i} \backslash C_{5o}$ | 0.75 | 0.54 | 0.59 | 0.77 | 0.80 | 0.74 | 0.75 | **0.89** |
| | $C_{1i} \backslash C_{6o}$ | 0.54 | 0.66 | 0.69 | 0.60 | 0.60 | 0.61 | 0.63 | **0.69** | $C_{2i} \backslash C_{6o}$ | 0.56 | 0.66 | 0.66 | 0.61 | 0.62 | 0.63 | 0.63 | **0.68** |
| | mean | 0.66 | 0.66 | 0.67 | 0.73 | 0.74 | 0.69 | 0.73 | **0.79** | - | 0.66 | 0.62 | 0.64 | 0.69 | 0.69 | 0.66 | 0.69 | **0.78** |
| AuPR (outlier) → | $C_{1i} \backslash C_{1o}$ | 0.35 | 0.35 | 0.35 | 0.36 | 0.37 | 0.38 | 0.38 | **0.41** | $C_{2i} \backslash C_{1o}$ | 0.44 | 0.41 | 0.41 | 0.60 | 0.59 | 0.55 | 0.53 | **0.62** |
| | $C_{1i} \backslash C_{2o}$ | 0.28 | 0.28 | 0.29 | 0.63 | 0.40 | 0.35 | 0.41 | **0.69** | $C_{2i} \backslash C_{2o}$ | 0.55 | 0.70 | 0.70 | 0.71 | 0.48 | 0.52 | 0.44 | **0.92** |
| | $C_{1i} \backslash C_{3o}$ | 0.27 | 0.21 | 0.42 | 0.64 | 0.35 | 0.38 | 0.37 | **0.81** | $C_{2i} \backslash C_{3o}$ | 0.41 | 0.26 | 0.29 | 0.61 | 0.48 | 0.51 | 0.53 | **0.65** |
| | $C_{1i} \backslash C_{4o}$ | 0.31 | 0.26 | 0.39 | 0.61 | 0.43 | 0.43 | 0.38 | **0.69** | $C_{2i} \backslash C_{4o}$ | 0.38 | 0.50 | 0.49 | 0.61 | 0.49 | 0.53 | 0.58 | **0.69** |
| | $C_{1i} \backslash C_{5o}$ | 0.29 | 0.21 | 0.41 | 0.73 | 0.52 | 0.31 | 0.28 | **0.80** | $C_{2i} \backslash C_{5o}$ | 0.43 | 0.48 | 0.47 | 0.75 | 0.70 | 0.71 | 0.80 | 0.80 |
| | $C_{1i} \backslash C_{6o}$ | 0.46 | 0.57 | 0.52 | 0.66 | 0.53 | 0.40 | 0.39 | **0.71** | $C_{2i} \backslash C_{6o}$ | 0.59 | 0.54 | 0.54 | 0.66 | 0.70 | 0.69 | 0.68 | **0.71** |
| | mean | 0.32 | 0.31 | 0.39 | 0.60 | 0.43 | 0.48 | 0.59 | **0.64** | - | 0.45 | 0.47 | 0.47 | 0.65 | 0.58 | 0.57 | 0.60 | **0.72** |

Table 11: Comparison of ENVISE with the baseline methods on IL class pairs $C_3$ and $C_4$ with every OL class on CIFAR-100. The ↓ indicates smaller value is better and ↑ indicates greater value is better.

| Metric | inlier \ outlier class | ED | CE | CS | ODIN | OECC | MCD | CAL | ENVISE | inlier \ outlier class | ED | CE | CS | ODIN | OECC | MCD | CAL | ENVISE |
|---|---|---|---|---|---|---|---|---|---|---|---|---|---|---|---|---|---|---|
| FPR (95% TPR) ↓ | $C_{3i} \backslash C_{1o}$ | 0.96 | 0.89 | 0.83 | 0.86 | 0.74 | 0.78 | 0.68 | **0.47** | $C_{4i} \backslash C_{1o}$ | 0.94 | 1.0 | 1.0 | 0.71 | 0.86 | 0.85 | 0.77 | **0.51** |
| | $C_{3i} \backslash C_{2o}$ | 0.92 | 0.87 | 0.87 | 0.76 | 0.82 | 0.82 | 0.79 | **0.57** | $C_{4i} \backslash C_{2o}$ | 0.93 | 0.96 | 0.96 | 0.73 | 0.89 | 0.87 | 0.67 | **0.59** |
| | $C_{3i} \backslash C_{3o}$ | 0.83 | 0.70 | 0.70 | 0.72 | 0.90 | 0.89 | 0.69 | **0.57** | $C_{4i} \backslash C_{3o}$ | 0.95 | 0.90 | 0.90 | 0.80 | 0.91 | 0.92 | 0.69 | **0.62** |
| | $C_{3i} \backslash C_{4o}$ | 0.94 | 0.92 | 0.94 | 0.92 | 0.92 | 0.90 | 0.77 | **0.73** | $C_{4i} \backslash C_{4o}$ | 0.91 | 0.65 | 0.66 | 0.66 | 0.97 | 0.94 | 0.87 | **0.61** |
| | $C_{3i} \backslash C_{5o}$ | 0.95 | 0.95 | 0.95 | 0.80 | 0.68 | 0.70 | 0.48 | **0.29** | $C_{4i} \backslash C_{5o}$ | 0.93 | 0.89 | 0.90 | 0.80 | 0.79 | 0.82 | 0.62 | **0.33** |
| | $C_{3i} \backslash C_{6o}$ | 0.92 | 0.85 | 0.85 | 0.87 | 0.86 | 0.87 | 0.69 | **0.65** | $C_{4i} \backslash C_{6o}$ | 0.92 | 0.87 | 0.87 | 0.74 | 0.92 | 0.90 | 0.70 | **0.61** |
| | mean | 0.92 | 0.87 | 0.86 | 0.80 | 0.81 | 0.83 | 0.72 | **0.55** | - | 0.94 | 0.93 | 0.93 | 0.78 | 0.88 | 0.88 | 0.73 | **0.57** |
| Detection error ↓ | $C_{3i} \backslash C_{1o}$ | 0.47 | 0.48 | 0.48 | 0.41 | 0.26 | 0.34 | 0.28 | **0.22** | $C_{4i} \backslash C_{1o}$ | 0.42 | 0.5 | 0.5 | 0.41 | 0.32 | 0.31 | 0.29 | **0.26** |
| | $C_{3i} \backslash C_{2o}$ | 0.42 | 0.44 | 0.43 | 0.40 | 0.33 | 0.33 | 0.31 | **0.30** | $C_{4i} \backslash C_{2o}$ | 0.45 | 0.5 | 0.5 | 0.44 | 0.36 | 0.34 | 0.39 | **0.30** |
| | $C_{3i} \backslash C_{3o}$ | 0.32 | 0.22 | 0.21 | 0.22 | 0.50 | 0.50 | 0.39 | **0.18** | $C_{4i} \backslash C_{3o}$ | 0.49 | 0.46 | 0.48 | 0.37 | 0.39 | 0.38 | 0.41 | **0.31** |
| | $C_{3i} \backslash C_{4o}$ | 0.38 | 0.48 | 0.48 | 0.36 | 0.41 | 0.38 | **0.29** | 0.32 | $C_{4i} \backslash C_{4o}$ | 0.42 | 0.27 | 0.28 | 0.27 | 0.45 | 0.42 | 0.42 | **0.26** |
| | $C_{3i} \backslash C_{5o}$ | 0.50 | 0.49 | 0.49 | 0.32 | 0.20 | 0.29 | 0.19 | **0.14** | $C_{4i} \backslash C_{5o}$ | 0.47 | 0.47 | 0.47 | 0.41 | 0.22 | 0.21 | 0.27 | **0.16** |
| | $C_{3i} \backslash C_{6o}$ | 0.38 | 0.41 | 0.40 | 0.33 | 0.35 | 0.36 | 0.37 | **0.32** | $C_{4i} \backslash C_{6o}$ | 0.42 | 0.41 | 0.41 | 0.40 | 0.37 | 0.34 | 0.33 | **0.31** |
| | mean | 0.42 | 0.42 | 0.42 | 0.35 | 0.32 | 0.34 | 0.31 | **0.25** | - | 0.44 | 0.44 | 0.45 | 0.39 | 0.34 | 0.33 | 0.35 | **0.27** |
| AuROC ↑ | $C_{3i} \backslash C_{1o}$ | 0.57 | 0.60 | 0.59 | 0.74 | 0.79 | 0.81 | 0.79 | **0.83** | $C_{4i} \backslash C_{1o}$ | 0.58 | 0.29 | 0.26 | 0.70 | 0.72 | 0.74 | 0.73 | **0.75** |
| | $C_{3i} \backslash C_{2o}$ | 0.59 | 0.46 | 0.47 | 0.74 | 0.72 | 0.73 | 0.69 | **0.77** | $C_{4i} \backslash C_{2o}$ | 0.55 | 0.38 | 0.38 | 0.60 | 0.65 | 0.70 | 0.73 | **0.78** |
| | $C_{3i} \backslash C_{3o}$ | 0.73 | 0.85 | 0.86 | 0.85 | 0.45 | 0.43 | 0.63 | **0.89** | $C_{4i} \backslash C_{3o}$ | 0.50 | 0.39 | 0.40 | 0.56 | 0.61 | 0.65 | 0.66 | **0.69** |
| | $C_{3i} \backslash C_{4o}$ | 0.59 | 0.41 | 0.42 | 0.79 | 0.60 | 0.64 | 0.75 | **0.84** | $C_{4i} \backslash C_{4o}$ | 0.71 | 0.84 | **0.85** | 0.84 | 0.66 | 0.65 | 0.67 | 0.82 |
| | $C_{3i} \backslash C_{5o}$ | 0.49 | 0.39 | 0.38 | 0.72 | 0.86 | 0.87 | 0.86 | **0.93** | $C_{4i} \backslash C_{5o}$ | 0.52 | 0.41 | 0.40 | 0.62 | 0.82 | 0.84 | 0.88 | **0.90** |
| | $C_{3i} \backslash C_{6o}$ | 0.64 | 0.59 | 0.59 | 0.73 | 0.68 | 0.67 | 0.69 | **0.75** | $C_{4i} \backslash C_{6o}$ | 0.59 | 0.53 | 0.53 | 0.78 | 0.66 | 0.69 | 0.75 | **0.78** |
| | mean | 0.54 | 0.42 | 0.42 | 0.35 | 0.32 | 0.34 | 0.31 | **0.84** | - | 0.57 | 0.44 | 0.45 | 0.39 | 0.34 | 0.33 | 0.35 | **0.78** |
| AuPR (inlier) ↑ | $C_{3i} \backslash C_{1o}$ | 0.55 | 0.42 | 0.40 | 0.73 | 0.70 | 0.69 | 0.66 | **0.74** | $C_{4i} \backslash C_{1o}$ | 0.68 | 0.43 | 0.44 | 0.71 | 0.69 | 0.69 | 0.68 | **0.73** |
| | $C_{3i} \backslash C_{2o}$ | 0.57 | 0.41 | 0.41 | 0.75 | 0.72 | 0.70 | 0.73 | **0.78** | $C_{4i} \backslash C_{2o}$ | 0.65 | 0.44 | 0.45 | 0.64 | 0.63 | 0.67 | 0.67 | **0.68** |
| | $C_{3i} \backslash C_{3o}$ | 0.75 | 0.85 | 0.86 | 0.85 | 0.45 | 0.43 | 0.63 | **0.91** | $C_{4i} \backslash C_{3o}$ | 0.62 | 0.45 | 0.44 | 0.62 | **0.69** | 0.63 | 0.66 | **0.69** |
| | $C_{3i} \backslash C_{4o}$ | 0.57 | 0.43 | 0.43 | 0.76 | 0.57 | 0.70 | 0.71 | **0.77** | $C_{4i} \backslash C_{4o}$ | 0.71 | 0.84 | **0.85** | 0.84 | 0.66 | 0.65 | 0.67 | 0.82 |
| | $C_{3i} \backslash C_{5o}$ | 0.50 | 0.39 | 0.38 | 0.72 | 0.86 | 0.89 | 0.86 | **0.91** | $C_{4i} \backslash C_{5o}$ | 0.62 | 0.53 | 0.53 | 0.74 | 0.88 | 0.81 | 0.83 | **0.92** |
| | $C_{3i} \backslash C_{6o}$ | 0.45 | 0.39 | 0.41 | 0.72 | 0.49 | 0.67 | 0.69 | **0.68** | $C_{4i} \backslash C_{6o}$ | 0.66 | 0.79 | 0.79 | 0.79 | 0.39 | 0.43 | 0.44 | **0.87** |
| | mean | 0.54 | 0.54 | 0.54 | 0.76 | 0.67 | 0.74 | 0.73 | **0.84** | - | 0.57 | 0.44 | 0.45 | 0.67 | 0.68 | 0.70 | 0.73 | **0.78** |
| AuPR (outlier) ↑ | $C_{3i} \backslash C_{1o}$ | 0.54 | 0.49 | 0.48 | 0.74 | 0.77 | 0.77 | 0.69 | **0.84** | $C_{4i} \backslash C_{1o}$ | 0.45 | 0.24 | 0.26 | 0.67 | 0.56 | 0.59 | 0.66 | **0.71** |
| | $C_{3i} \backslash C_{2o}$ | 0.57 | 0.43 | 0.43 | 0.73 | 0.69 | 0.70 | 0.68 | **0.76** | $C_{4i} \backslash C_{2o}$ | 0.42 | 0.25 | 0.24 | 0.60 | 0.63 | 0.66 | **0.67** | **0.67** |
| | $C_{3i} \backslash C_{3o}$ | 0.73 | 0.86 | 0.87 | 0.86 | 0.46 | 0.45 | 0.56 | **0.93** | $C_{4i} \backslash C_{3o}$ | 0.39 | 0.25 | 0.25 | 0.56 | 0.47 | 0.59 | 0.62 | **0.68** |
| | $C_{3i} \backslash C_{4o}$ | 0.56 | 0.49 | 0.48 | 0.72 | 0.56 | 0.69 | 0.69 | **0.74** | $C_{4i} \backslash C_{4o}$ | 0.46 | 0.72 | 0.70 | 0.72 | 0.40 | 0.46 | 0.49 | 0.77 |
| | $C_{3i} \backslash C_{5o}$ | 0.50 | 0.45 | 0.44 | 0.72 | 0.83 | 0.82 | 0.86 | **0.92** | $C_{4i} \backslash C_{5o}$ | 0.41 | 0.38 | 0.37 | 0.69 | 0.66 | 0.69 | 0.77 | **0.86** |
| | $C_{3i} \backslash C_{6o}$ | 0.75 | 0.77 | 0.77 | 0.77 | 0.78 | 0.77 | 0.72 | **0.82** | $C_{4i} \backslash C_{6o}$ | 0.61 | 0.44 | 0.44 | 0.73 | 0.66 | 0.69 | 0.71 | **0.78** |
| | mean | 0.60 | 0.57 | 0.57 | 0.75 | 0.71 | 0.73 | 0.66 | **0.83** | - | 0.47 | 0.37 | 0.37 | 0.65 | 0.57 | 0.62 | 0.65 | **0.72** |

Table 12: Comparison of ENVISE with the baseline methods on IL class pairs $C_5$ and $C_6$ with every OL class on CIFAR-100. The ↓ indicates smaller value is better and ↑ indicates greater value is better.

| Metric | inlier\outlier class | ED | CE | CS | ODIN | OECC | MCD | CAL | ENVISE | inlier\outlier class | ED | CE | CS | ODIN | OECC | MCD | CAL | ENVISE |
|---|---|---|---|---|---|---|---|---|---|---|---|---|---|---|---|---|---|---|
| FPR (95% TPR) ↓ | $C_{5i}\backslash C_{1o}$ | 0.95 | 0.88 | 0.89 | 0.77 | 0.89 | 0.92 | 0.73 | **0.62** | $C_{6i}\backslash C_{1o}$ | 0.93 | 0.72 | 0.73 | 0.73 | 0.82 | 0.79 | 0.71 | **0.63** |
| | $C_{5i}\backslash C_{2o}$ | 0.94 | 1.0 | 1.0 | 0.76 | 0.84 | 0.87 | 0.81 | **0.72** | $C_{6i}\backslash C_{2o}$ | 0.93 | 0.79 | 0.79 | 0.72 | 0.85 | 0.80 | 0.66 | **0.55** |
| | $C_{5i}\backslash C_{3o}$ | 0.93 | 0.94 | 0.94 | 0.74 | 0.90 | 0.91 | 0.72 | **0.69** | $C_{6i}\backslash C_{3o}$ | 0.95 | 0.81 | 0.82 | 0.81 | 0.87 | 0.86 | 0.64 | **0.58** |
| | $C_{5i}\backslash C_{4o}$ | 0.95 | 1.0 | 1.0 | **0.78** | 0.94 | 0.94 | 0.81 | 0.92 | $C_{6i}\backslash C_{4o}$ | 0.95 | 0.87 | 0.87 | 0.77 | 0.87 | 0.86 | 0.69 | **0.51** |
| | $C_{5i}\backslash C_{5o}$ | 0.94 | 0.92 | 0.93 | 0.92 | 0.90 | 0.95 | 0.82 | **0.76** | $C_{6i}\backslash C_{5o}$ | 0.95 | 0.73 | 0.74 | 0.74 | 0.76 | 0.73 | 0.53 | **0.37** |
| | $C_{5i}\backslash C_{6o}$ | 0.94 | 1.0 | 1.0 | 0.74 | 0.92 | 0.93 | 0.88 | **0.59** | $C_{6i}\backslash C_{6o}$ | 0.83 | 0.69 | 0.68 | 0.66 | 0.89 | 0.95 | 0.63 | **0.49** |
| | mean | 0.94 | 0.96 | 0.96 | 0.79 | 0.90 | 0.92 | 0.82 | **0.72** | - | 0.92 | 0.76 | 0.76 | 0.75 | 0.84 | 0.83 | 0.69 | **0.52** |
| Detection error ↓ | $C_{5i}\backslash C_{1o}$ | 0.42 | 0.43 | 0.44 | 0.42 | 0.34 | 0.34 | 0.33 | **0.30** | $C_{6i}\backslash C_{1o}$ | 0.43 | 0.36 | 0.36 | 0.41 | 0.37 | 0.39 | **0.32** | **0.32** |
| | $C_{5i}\backslash C_{2o}$ | 0.45 | 0.50 | 0.50 | 0.39 | 0.35 | 0.34 | 0.41 | **0.33** | $C_{6i}\backslash C_{2o}$ | 0.44 | 0.42 | 0.42 | 0.40 | 0.30 | 0.32 | 0.30 | **0.29** |
| | $C_{5i}\backslash C_{3o}$ | 0.40 | 0.43 | 0.44 | 0.38 | 0.38 | 0.37 | 0.41 | **0.36** | $C_{6i}\backslash C_{3o}$ | 0.46 | 0.41 | 0.40 | 0.37 | 0.36 | 0.36 | 0.32 | **0.30** |
| | $C_{5i}\backslash C_{4o}$ | 0.43 | 0.50 | 0.50 | **0.39** | 0.43 | 0.44 | **0.39** | 0.43 | $C_{6i}\backslash C_{4o}$ | 0.49 | 0.45 | 0.44 | 0.39 | 0.30 | 0.32 | 0.28 | **0.26** |
| | $C_{5i}\backslash C_{5o}$ | 0.41 | 0.39 | 0.39 | 0.39 | 0.46 | 0.44 | 0.44 | **0.29** | $C_{6i}\backslash C_{5o}$ | 0.43 | 0.37 | 0.37 | 0.39 | 0.19 | 0.21 | 0.28 | **0.18** |
| | $C_{5i}\backslash C_{6o}$ | 0.40 | 0.50 | 0.50 | 0.40 | 0.38 | 0.38 | 0.34 | **0.26** | $C_{6i}\backslash C_{6o}$ | 0.29 | 0.22 | 0.22 | 0.22 | 0.43 | 0.40 | 0.42 | **0.16** |
| | mean | 0.43 | 0.47 | 0.47 | 0.40 | 0.39 | 0.39 | 0.39 | **0.33** | - | 0.43 | 0.37 | 0.37 | 0.37 | 0.31 | 0.35 | 0.32 | **0.25** |
| AuROC ↑ | $C_{5i}\backslash C_{1o}$ | 0.59 | 0.42 | 0.43 | 0.64 | 0.68 | 0.68 | 0.58 | **0.69** | $C_{6i}\backslash C_{1o}$ | 0.58 | 0.65 | 0.65 | 0.65 | 0.77 | 0.76 | 0.79 | **0.83** |
| | $C_{5i}\backslash C_{2o}$ | 0.56 | 0.40 | 0.41 | 0.69 | 0.68 | 0.69 | 0.72 | **0.76** | $C_{6i}\backslash C_{2o}$ | 0.56 | 0.53 | 0.54 | 0.65 | 0.64 | 0.63 | 0.66 | **0.71** |
| | $C_{5i}\backslash C_{3o}$ | 0.61 | 0.36 | 0.36 | 0.68 | 0.63 | 0.64 | 0.62 | **0.71** | $C_{6i}\backslash C_{3o}$ | 0.55 | 0.60 | 0.60 | 0.67 | 0.68 | 0.67 | 0.70 | **0.75** |
| | $C_{5i}\backslash C_{4o}$ | 0.58 | 0.36 | 0.35 | 0.57 | 0.57 | 0.56 | 0.63 | **0.69** | $C_{6i}\backslash C_{4o}$ | 0.50 | 0.53 | 0.53 | 0.62 | 0.75 | 0.72 | 0.69 | **0.76** |
| | $C_{5i}\backslash C_{5o}$ | 0.59 | 0.62 | 0.62 | 0.62 | 0.50 | 0.56 | 0.54 | **0.79** | $C_{6i}\backslash C_{5o}$ | 0.59 | 0.66 | 0.66 | 0.66 | 0.86 | 0.86 | **0.88** | **0.88** |
| | $C_{5i}\backslash C_{6o}$ | 0.40 | 0.47 | 0.47 | **0.77** | 0.63 | 0.64 | 0.69 | 0.76 | $C_{6i}\backslash C_{6o}$ | 0.78 | 0.85 | 0.84 | 0.85 | 0.57 | 0.60 | 0.60 | **0.91** |
| | mean | 0.56 | 0.43 | 0.44 | 0.66 | 0.63 | 0.63 | 0.64 | **0.73** | - | 0.58 | 0.64 | 0.64 | 0.68 | 0.73 | 0.67 | 0.73 | **0.81** |
| AuPR (inlier) ↑ | $C_{5i}\backslash C_{1o}$ | 0.87 | 0.71 | 0.71 | 0.75 | 0.70 | 0.70 | 0.79 | **0.88** | $C_{6i}\backslash C_{1o}$ | 0.55 | 0.57 | 0.57 | 0.64 | 0.72 | 0.73 | 0.69 | **0.74** |
| | $C_{5i}\backslash C_{2o}$ | 0.86 | 0.70 | 0.71 | 0.69 | 0.69 | 0.70 | 0.66 | **0.87** | $C_{6i}\backslash C_{2o}$ | 0.54 | 0.49 | 0.50 | 0.66 | 0.71 | 0.70 | 0.73 | **0.77** |
| | $C_{5i}\backslash C_{3o}$ | 0.88 | 0.76 | 0.77 | 0.78 | 0.68 | 0.79 | 0.80 | **0.85** | $C_{6i}\backslash C_{3o}$ | 0.52 | 0.54 | 0.55 | 0.67 | 0.64 | 0.62 | 0.71 | **0.75** |
| | $C_{5i}\backslash C_{4o}$ | **0.87** | 0.69 | 0.69 | 0.72 | 0.75 | 0.75 | 0.71 | 0.80 | $C_{6i}\backslash C_{4o}$ | 0.49 | 0.50 | 0.50 | 0.61 | 0.65 | 0.60 | **0.66** | 0.66 |
| | $C_{5i}\backslash C_{5o}$ | 0.87 | 0.90 | 0.90 | 0.90 | 0.85 | 0.87 | 0.85 | **0.95** | $C_{6i}\backslash C_{5o}$ | 0.58 | 0.60 | 0.60 | 0.66 | 0.79 | 0.82 | 0.79 | **0.84** |
| | $C_{5i}\backslash C_{6o}$ | 0.78 | 0.59 | 0.59 | 0.63 | 0.69 | 0.77 | 0.80 | **0.85** | $C_{6i}\backslash C_{6o}$ | 0.56 | 0.78 | 0.79 | 0.78 | 0.58 | 0.26 | 0.46 | **0.87** |
| | mean | 0.85 | 0.72 | 0.73 | 0.74 | 0.73 | 0.76 | 0.77 | **0.87** | - | 0.55 | 0.58 | 0.59 | 0.67 | 0.69 | 0.57 | 0.56 | **0.77** |
| AuPR (outlier) ↑ | $C_{5i}\backslash C_{1o}$ | 0.19 | 0.20 | 0.21 | 0.50 | 0.56 | 0.56 | 0.62 | **0.63** | $C_{6i}\backslash C_{1o}$ | 0.55 | 0.70 | 0.71 | 0.64 | 0.72 | 0.73 | 0.69 | **0.74** |
| | $C_{5i}\backslash C_{2o}$ | 0.19 | 0.19 | 0.20 | **0.77** | 0.40 | 0.60 | 0.68 | 0.62 | $C_{6i}\backslash C_{2o}$ | 0.54 | 0.62 | 0.62 | 0.64 | 0.71 | 0.71 | 0.69 | **0.78** |
| | $C_{5i}\backslash C_{3o}$ | 0.22 | 0.22 | 0.23 | 0.37 | 0.43 | 0.45 | 0.49 | **0.56** | $C_{6i}\backslash C_{3o}$ | 0.52 | 0.62 | 0.64 | 0.66 | 0.66 | 0.65 | 0.71 | **0.75** |
| | $C_{5i}\backslash C_{4o}$ | 0.19 | 0.30 | 0.30 | 0.43 | 0.31 | **0.49** | **0.49** | **0.49** | $C_{6i}\backslash C_{4o}$ | 0.49 | 0.57 | 0.57 | 0.66 | 0.70 | 0.69 | 0.73 | **0.81** |
| | $C_{5i}\backslash C_{5o}$ | 0.23 | 0.21 | 0.21 | 0.21 | 0.16 | 0.19 | 0.19 | **0.42** | $C_{6i}\backslash C_{5o}$ | 0.53 | 0.69 | 0.69 | 0.65 | 0.81 | 0.81 | 0.80 | **0.90** |
| | $C_{5i}\backslash C_{6o}$ | 0.34 | 0.29 | 0.29 | 0.41 | 0.47 | 0.48 | 0.44 | **0.50** | $C_{6i}\backslash C_{6o}$ | 0.86 | 0.90 | 0.89 | 0.90 | 0.70 | 0.74 | 0.74 | **0.97** |
| | mean | 0.22 | 0.24 | 0.24 | 0.45 | 0.40 | 0.37 | 0.48 | **0.54** | - | 0.57 | 0.68 | 0.69 | 0.69 | 0.73 | 0.70 | 0.73 | **0.82** |

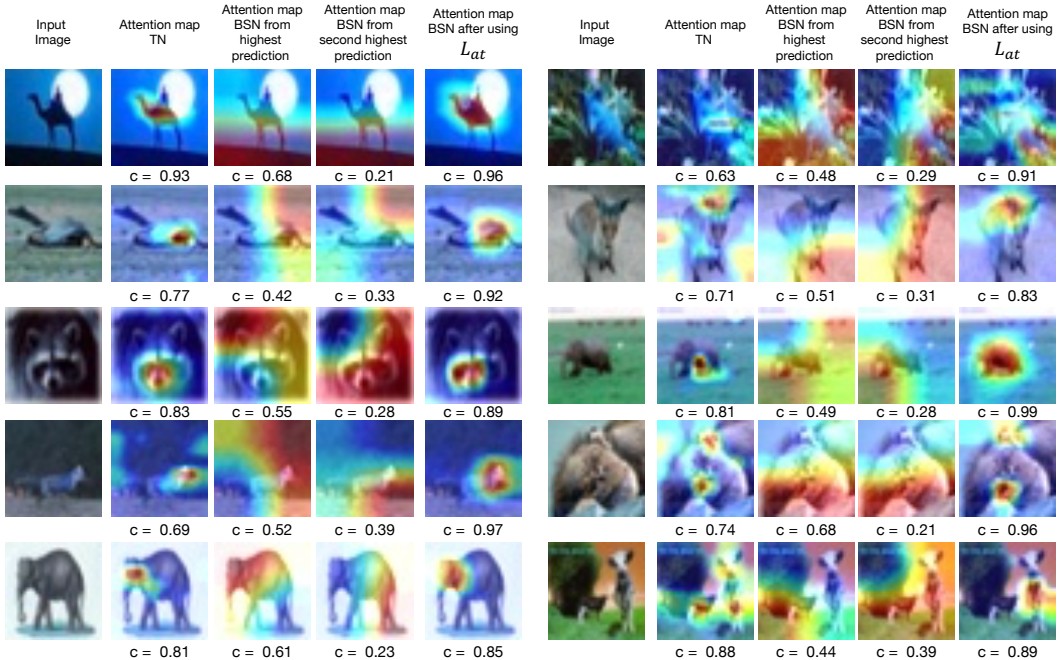

Figure 11: The attention map visualization representing the regions of the image that the network focuses on for classification. The confidence of correct classification is reported below each image

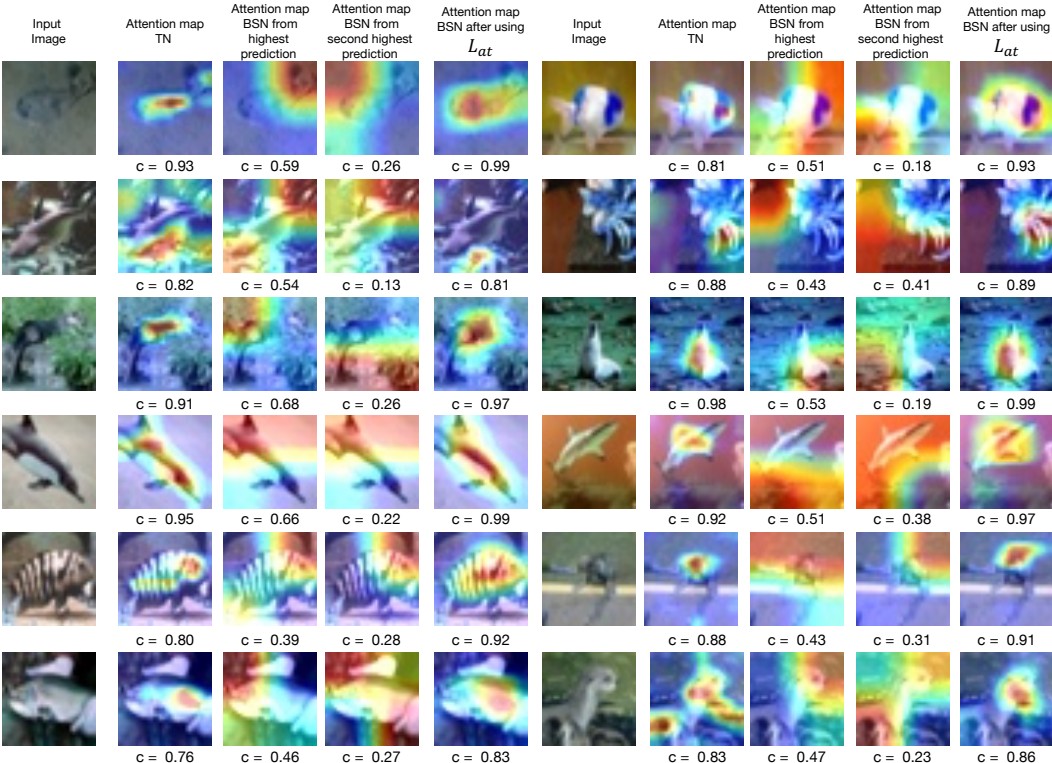

Figure 12: The attention map visualization representing the regions of the image that the network focuses on for classification. The confidence of correct classification is reported below each image

