# OpenReview forum: "Enhancing Visual Representations for Efficient Object Recognition during Online Distillation"
_ICLR.cc/2021/Conference — Reject_

### Official Review · AnonReviewer1 · 2020-10-24
**Knowledge distillation for efficient object recognition at scale with attention supervision. The methodology is not well motivated.**

**Rating:** 4
**Confidence:** 4

**Review:**

A knowledge distillation framework is proposed for efficient object recognition. In this framework, the teacher network (TN) performs high accuracy prediction while two student networks (SN) mimic the prediction from TN. The first SN learns from TN while the second SN is a binarized form (BSN) of the first SN.  The design is made for online inference is that the BSN first recognizes the image, leaving the rare category objects to be recognized by the TN. Furthermore, an attention supervision scheme is proposed to enhance the CNN prediction by focusing on meaningful image content.  The proposed method has been validated on CIFAR-100 and Tiny-Imagenet.

While the distillation for fast recognition and attention supervisions sound interesting, there are a few issues to make the current manuscript not convincing on the contribution side. The details are listed in the following:

1. The major issue is on the motivation side of this framework design. In the paper, a BSN and a TN are both adopted to recognize objects. The reason claimed is for the efficient recognition for rare category objects. However, there is no motivation for why distillation can indeed solve this problem.  This reviewer agrees that objects from category distribute unevenly and class imbalance occurs during training. How class imbalance correlates to knowledge distillation is not clear. A common strategy is to adopt a focal-loss-based loss function to reduce contributions from easy samples. Suppose we want to use two CNNs for cascaded recognition, which is the core idea in this paper, the typical choice is to collect ordinary category objects for the first stage training and rare objects for the second stage training. A pure distillation from the TN does not ensure the differentiation of ordinary and rare categories.

2. The claim of online distillation is weird. Normally the CNN is trained offline and there is no sign of online model training in the manuscript. As online distillation appears everywhere in the manuscript, this reviewer doubts whether this claim is suitable for illustrating the CNN training process.

3. The attention scheme is from Grad-CAM, which back-propagates CNN prediction to formulate attention supervision terms. The number of categories indicates the number of back-propagation during one training iteration. This computational complexity is tremendous compared to the fast convergence claim of BSN. There shall be some analysis on how to compute the attention maps efficiently in practice.

4. In the experiments, the comparison shall be made to the sota object recognition methods (ResNet, EfficientNet based methods) besides outlier detection methods, As the proposed method focuses on efficient object recogntion.

---

> ### Author Response · Authors · 2020-11-16
> **Thank you for your feedback and answers to your queries (1/2)**
>
> We thank R1 for the valuable feedback. We humbly ask R1 to read our response below and if agreed, to kindly consider raising the rating.
>
> **[class-imbalance during inference and not during training]** We would like to explain our motivation for using online distillation to deal with class-imbalance in the deployed scenario
> - *The adaptive learning of the BSN is during the deployed state i.e. during inference and not during the training phase*. We mention that our motivation stems from the observation that in the deployed scenario, the probability of occurrence of all classes is not the same and only a subset of classes occur frequently. Hence, in our problem setting, *class-imbalance occurs during inference* and not during training.
> - In the deployed scenario i.e. inference, *we do not have access to ground truth labels* to adaptively learn the BSN on the frequently occurring classes. We employ a high performance, general-purpose oracle network as our TN that is capable of generating hard pseudo-labels to “distill” knowledge of only the frequently occurring classes into the BSN. Although the TN has knowledge of all the possible classes, *only a subset of its vast knowledge may be useful in the deployed state* where a handful of classes occur frequently, and the rest are rare. Thus, we distill knowledge of frequently occurring classes to learn a low-capacity BSN using hard pseudo-labels from the TN, capable of *classifying these frequently occurring classes with high accuracy and efficiency* (Section 4 - “Gradually changing the deployed scenario”)
>  - The suggestion of the reviewer for using focal-loss-based loss function and cascaded CNN architectures is noted when class-imbalance occurs during training. We also acknowledge that existing works [3,4] employ knowledge distillation to address class-imbalance only during training. However, in our work, we address *class-imbalance during the deployed scenario i.e. during inference* and hence use online distillation.
>
> **[online distillation]** In our response to R4, we mention that, the term online distillation is inspired from Mullapudi et al. (2019), who distill information from teacher to the student network on a live input stream as the new data (video frames in their case) arrive into the network i.e. in an online fashion.
> - Mullapudi et al. (2019) claim that they perform online distillation inspired from the observation that, most real-world video cameras capture distributions that continuously evolve over time and are different from the training distribution.
> - [5] claim that they perform online test-time training inspired from the observation that in many real-world applications training and test data are drawn from different distributions.
> - Similar to our setting,  [6, 7] claim that during inference, prior probability of occurrence of classes are different (i.e. class imbalance during inference) and estimate test-time priors by  adapting the predicted class scores and MLE and MAP estimation using a Dirichlet hyperprior respectively.
> Thus, the difference between training and test statistics (distribution shifts, varying prior probabilities of classes  etc.) has led to existing works to rely on online learning techniques which is different from conventional offline CNN training. We use online distillation based on our observation that *class-imbalance occurs during inference (deployed scenario)* and employ the BSN to classify frequently occurring classes with high accuracy and efficiency (Figure 3 and Figure 5).
>
> **[computation overhead of attention map]** The attention map computed using Grad-CAM does not produce any computation overhead. As mentioned in the 2nd para of page 5, following Grad-CAM, for a given image, the attention map is computed by backpropagating the gradients from the one-hot prediction to the last convolution layer (and not through the entire network). Furthermore, *we are not computing attention maps for all classes*, but only the attention map from the most probable class of TN and BSN, and second most probable class of BSN. Since $L_{at}$ is formulated using only 3 attention maps, *for a given image, we only compute 3 attention maps* which do not lead to high computational cost.
>
> **[comparison with SOTA efficiency based methods]** In Section 4, under “Comparison with SOTA outlier detection methods” we mention that SOTA model compression techniques (Frankle & Carbin (2019)) do not deal with processing images from the input stream with varying prior class probabilities. Hence, direct comparison with these methods is not meaningful since their objectives are different from those of ENVISE. R3 also acknowledges in their strength that our “Experimental evaluation is very thorough”.

---

> > ### Author Response · Authors · 2020-11-16
> > **References (2/2)**
> >
> > [3] Chen, G., Choi, W., Yu, X., Han, T. and Chandraker, M., Learning efficient object detection models with knowledge distillation. NeurIPS 2017
> >
> > [4] Xiang, L., Ding, G. and Han, J., Learning from multiple experts: Self-paced knowledge distillation for long-tailed classification. ECCV 2020
> >
> > [5] Sun, Y., Wang, X., Liu, Z., Miller, J., Efros, A.A. and Hardt, M.. Test-time training with self-supervision for generalization under distribution shifts. ICML 2020
> >
> > [6] Royer, A. and Lampert, C.H., Classifier adaptation at prediction time. CVPR 2015
> >
> > [7] Sulc, M. and Matas, J., Improving CNN classifiers by estimating test-time priors. ICCV-W 2019

---

### Official Review · AnonReviewer2 · 2020-10-27
**Interesting paper but needs some clarification**

**Rating:** 5
**Confidence:** 3

**Review:**

The authors tackle the problem of efficient object recognition and outlier detection using online distillation. They propose to complement the standard distillation loss with a triplet loss using attention maps from the teacher to help the student focusing on the relevant part of the input images. They also provide a large set of experiments to validate the proposed approach. The paper is overall well written.

I have two main concerns about the paper:
- If my understanding is correct, the *hard-positive* samples used to compute the attention triplet loss are selected from the most probable class assigned by the TN while the *hard-negative* are using the second most probable class. Why is it desired for these first and second most probable categories to have different attention maps? Intuitively, if they are semantically similar it would make sense to focus on the same regions of the image to be able to distinguish them from each other.
- Even though discussed in section A.3, I think that the claim "BSN is adaptively trained only on IL class images without any knowledge of OL class images" is incorrect (first paragraph of page 3). Since the TN is used as supervision and as a guide for the SN representation, a lot of knowledge about the OL classes can still be transmitted to the SN.

And a few questions:
- How does the SN perform when directly trained on all classes? It would give a tight lower bound to the performance of ENVISE, since some architectures used for the SN are already very competitive (especially Resnet 18 and 50).
- It is surprising to see the BSN outperform the RvSN. Is there any reason for which the real-valued network can't converge to the same level of performance as the one achieved by the BSN?
- Since table 5 shows that similar performances can be reached using a Resnet-18, why using a Densnet-201 for most of the experiments? It seems that it can improve the performance in terms of GiE but would just increase the overall inference latency of ENVISE.


Minor remark:
- Legends of fig 2b, 3 and 4 are too small.

---

> ### Author Response · Authors · 2020-11-13
> **Thank you for your feedback and answers to your queries**
>
> We thank R2 for the valuable feedback. We humbly ask R2 to read our response below and if agreed, to kindly consider raising the rating.
>
> **[semantically similar attention map]** This is an interesting point, and we would like to further explain why we feel our approach is correct.
> - Firstly, the most probable class is not always assigned by the TN. Since the BSN is trained using *hard pseudo-labels* from the TN, only when the BSN misclassifies an image, we assign the most probable class using the TN (second para of page 5) to compute hard positive attention map.
> - Secondly, please consider the attention map from the most probable prediction (correct prediction) as positive hypothesis and that from second most probable class (incorrect prediction) as negative hypothesis. We know from  Grad-CAM (Selvaraju et al. (2017)) that the *attention map highlights the regions of the image responsible for the classifier’s prediction*. Hence, the positive and negative hypothesis should be semantically different from each other since they do not represent the same object in the image. For e.g. from 1st row in Figure 2(a), the most  probable class predicted by BSN is *camel* and second most probable class predicted is *leopard*. We expect that the attention maps corresponding to these predictions should be semantically different since the network’s activations are different for camel and leopard class. Furthermore, *the leopard and camel classes are not semantically similar and thus, their attention maps should also be different*. However, we observe that the attention maps from camel and leopard class of the BSN highlight the same region of the image, which signifies that the BSN learns incorrect representations. Thus, to make sure that the BSN learns the correct representations, we propose $L_{at}$ which ensures that positive and negative hypothesis are semantically different and that the positive hypothesis is similar to the attention map of the TN (Figure 2(a)).
>
> **[OL class knowledge]** We also initially had concerns over whether knowledge of the OL classes was being transferred to the SN from the TN, but after some experimental analysis, *we find this not to be the case*.  In prior knowledge distillation works  [Hinton et al. (2015), 1, 2], the final layer of the TN is distilled into the SN i.e. soft-targets which consists of knowledge about every class. However, in ENVISE, we mentioned in Section 3 under “Knowledge transfer from teacher to student network” that we *only use hard pseudo-label* from the TN and not the output of the final layer (i.e. soft-targets). In our formulation of cross-entropy loss $L_{d}$, we specifically mention that we use the pseudo-labels generated by the TN. Furthermore, the attention map from Grad-CAM is also computed using the *one-hot label of the predicted class* (second para of page 5). As noted by the reviewer, we show in Figure 9 in Section A.3 that the entropy of OL class of the BSN is high during online distillation. This validates our claim that *we do not transfer any OL class information during online distillation*.
>
> **[SN trained on all classes]** The yellow line in Figure 3 illustrates the convergence analysis of SN (in terms of accuracy) which has been pretrained on all classes. We will add the performance of SN when trained on all classes, in terms of FPR (at 95\% TPR), outlier detection error, AuROC, AuPR and GiE in Appendix A.4.
>
> **[RvSN trained for more epochs]** In *Section 4 under “Gradually changing the deployed scenario”*, we mention that we employ the BSN as our SN due to its ability to converge faster than RvSN. We empirically observed that if we increase the number of epochs from 10 to 50 for online distillation, the RvSN and BSN have similar performance in terms of accuracy and in some cases the RvSN outperforms the BSN. However, in this work, we focus on improving the efficiency of ENVISE which requires us to choose a SN that converges to the performance of the TN using fewer epochs. This would improve the GiE resulting in higher efficiency of the overall system.
>
> **[Experiments with resnet-18]** We show the comparison of performance of ENVISE  in terms of OL detection with Resnet-18 and DenseNet-201 as our TN in Table 5. Since our experimental analysis is huge, considering Table 3, Table 9 - 12, we will update our experiments with Resnet-18 as TN in the camera ready version as per your suggestion.
>
>
> We thank you for taking the time to read our response. We will be happy to answer your questions if you think that they have not been solved.
>
> [1] Zhu, X. and Gong, S.. Knowledge distillation by on-the-fly native ensemble. NeurIPS, 2018
>
> [2] Chen, D., Mei, J.P., Wang, C., Feng, Y. and Chen, C.. Online Knowledge Distillation with Diverse Peers. AAAI, 2020.

---

### Official Review · AnonReviewer3 · 2020-10-28
**Interesting use of triplet loss in distillation, paper organization and writing should be improved.**

**Rating:** 5
**Confidence:** 4

**Review:**

The proposed work trains a teacher-student network using an online distillation paradigm. The student is a binarized network (BSN) trained to be accurate on frequent classes. An attention triplet loss is employed to improve the accuracy of the BSN and its ability to detect outlier classes. Faster convergency of BSN vs Real Valued Student network is claimed. A new metric to evaluate the actual gain in network efficiency is proposed.

Strengths
- The idea of using the attention triplet loss to improve the quality of attention maps of the BSN and thus increasing BSN accuracy is interesting
- Experimental evaluation is very thorough

Weaknesses
The main weakness of this work lies in the presentation and organization of the paper.
- It is not clear how outlier detection is performed and why it is needed for the approach to correctly function.
- It is not clear why at inference time false outliers are then processed by the teacher network. Is this because of the online setting? The setting should be better specified.
- Sentence "In conventional knowledge distillation, the attention map of the BSN can focus on the background
even when the attention map of the TN emphasizes the semantically meaningful regions of the image.", should be followed by one (possibly more) citations supporting this claim.
- Lemma 3.1 should be followed by a sketch proof, being one of the main contribution, with the full proof in the appendix (as it is already).

Related work missing:
Existing literature binarizing CNN in distillation exists:
[a]Distillation Guided Residual Learning for Binary Convolutional Neural Networks, 2018
Triplet losses are used in distillation:
Triplet Loss for Knowledge Distillation, 2017 (arxiv), IJCNN(2020)


The main reason for my score regards the overall presentation of the paper. The main two contributions are the faster convergence and the use of triplet attention loss. The lemma should have been better highlighted with a sketch proof or at least some intuition so that readers not willing to sift through the appendix could get a grasp of it. The triplet attention loss should be better framed in the introduction (see above). Finally some of the mechanisms of the approach are unclear (how to get the OL score, why the TN must evaluate fp of the BSN) and so on.

---

> ### Author Response · Authors · 2020-11-12
> **Thank you for your feedback and answers to your queries**
>
> We thank R3 for the valuable feedback. We humbly ask R3 to read our response below and if agreed, to kindly consider raising the rating.
>
> **[outlier detection]** In *Section 4, under “Training and evaluation”*, we mention that for each image from the input stream during inference (comprising of IL and OL classes), following Hendrycks & Gimpel (2016), we compute the confidence of prediction from the softmax probability of the predicted class. If the confidence is low, we treat the image as an OL class and direct it to the TN for classification.
> In ENVISE, during inference, the BSN classifies the frequently occurring classes (IL class) with high accuracy (Figure 3) faster than the TN. Since the BSN is trained only on the IL classes, it would misclassify the OL class present in the image stream. *It is important to note that the OL classes are outliers with respect to the BSN only, but are known to the TN (2nd para of introduction).* Misclassifying an OL class would lead to a drop in the performance of the overall system thereby resulting in a poor efficiency (GiE). Thus, without sacrificing performance of ENVISE, we detect  OL classes unknown to the BSN and direct it to the TN for classification. This ensures that the overall accuracy of ENVISE is bounded by the performance of the TN (Table 6 in Appendix A.3).
>
>
> **[outliers processed by TN]** As mentioned in the 2nd para of introduction, the *OL classes are outliers with respect to the BSN only, but are known to the TN* . From  point a) since the BSN is trained on frequently occurring IL classes, it would misclassify a rare (outlier) class present in the image stream. To maintain the overall accuracy of ENVISE during inference, we direct the OL class rejected by the BSN to the TN. This ensures that the overall accuracy of ENVISE is not degraded, but is as close as possible to the performance of the TN. The idea of directing the rare OL class to the TN is further illustrated in *Appendix A.3 under “Trade-off between accuracy and GiE in ENVISE”.*
>
>
> **[citations for attention map]** As shown in Figure 2 (a), when the BSN is trained only using $L_{d}$,  *we have observed* that the attention map of the BSN may focus on the background even when the attention map of the TN emphasizes the semantically meaningful regions of the image. However, we acknowledge that the language used in 3rd para of introduction could cause confusion, regarding it as claims from existing works. We shall correct the current sentence as our empirical observation.
>
> **[sketch proof]** We have added the sketch proof in Lemma 3.1 as per your suggestion.
>
> **[add references]** We will also cite  the existing works on Distillation Guided Residual Learning for Binary Convolutional Neural Networks, 2018: Triplet Loss for Knowledge Distillation, 2017 in Related Works section.
>
>
> We thank you for taking the time to read our response. We will be happy to answer your questions if you think that they have not been solved.

---

### Official Review · AnonReviewer4 · 2020-10-30
**This paper proposes a knowledge distillation framework to achieve efficient object recognition when adapting a large model to a new scenario. It features in an adaption module to learn a binary student network and an attention triplet loss to guide the student network to focuse on semantic information. It mimics the data adaptation scenario by splititing the datasets CIFAR-100 and Tiny-Imagenet into inliner and outliner classes and shows improvement over the baseline on multiple metrics.**

**Rating:** 4
**Confidence:** 5

**Review:**

- The idea is interesting to focus on the frequent labels by distilling a special binary network. But the technical details are not clearly presented and important experiments are missing. Overall, this paper does not reach the acceptance threshold.

Detailed Comments:

- Why is this method an 'online' distillation considering that the teacher network is pretrained and fixed? How do you define offeline distillation and online distillation?
- "The OL detector uses the softmax output of the BSN..." How is the softmax output is used? Do you have a separate class named outlier apart from all the object categories? It should be made clear.
- Why does BSN converge faster than RvSN considering that BSN is a binarized version of RvSN? What is the retionale behind this?
- In the experimental part, only comparison to baseline methods are provided, but not to state-of-the-art efficient object recognition methods in the literature. Various methods have been proposed recently to improve the efficiency of an object recognition network, such as distillation, pruning, tensor decomposition, quatization. There should be comparison to these methods as well as baselines. Besides, the performance on Imagenet (not only tiny imagenet) is a common practice in the literature, but is missing in this paper.

---

> ### Author Response · Authors · 2020-11-12
> **Thank you for your feedback and answers to your queries**
>
> We thank R4 for the valuable feedback. We humbly ask R4 to read our response below and if agreed, to kindly consider raising the rating.
>
> **[online distillation]** We acknowledge that some existing knowledge distillation works [1,2] define online distillation as a one-phase learning procedure without using a pre-trained teacher model. However, our definition of the term *online distillation* is inspired by Mullapudi et al. (2019) who distill information from teacher to the student network on a live input stream as the new data (video frames in their case) arrive into the network i.e. in an online fashion.
> In [1,2], the student network is trained with the assumption that prior probability of occurrence of each class in the training data is equal, which is different from ENVISE.  In ENVISE, we assume that in the deployed scenario, some classes will occur more frequently than others and therefore the prior probability of concurrence of these classes vary. Hence, we adapt the BSN on the live stream of images as it arrives into ENVISE (after deployment) and term this learning process as online distillation.
>
>
> **[outlier detection]** The term IL and OL are interchangeably used to denote frequent and rare classes that are presented to the BSN. In *Section 4, under “Training and evaluation”*, we mention that for each image from the input stream  (comprising of IL and OL classes), following Hendrycks & Gimpel (2016), we compute the confidence of prediction using  the softmax probabilities produced by the BSN. If the confidence is low, we treat the image as an OL class and direct  it to the TN for classification.
> Hence, we do not have a predefined separate outlier class, but similar to Hendrycks & Gimpel (2016), we compute the confidence from the BSN’s softmax of the predicted class to detect rare classes as outliers .
>
>
> **[faster convergence of BSN]** We provide the complete proof of why the BSN converges faster than the real-valued SN in Appendix A.1. We have added a sketch proof of Lemma 3.1 as per the suggestion of R3.
>
> **[comparison with SOTA compression methods]** In *Section 4, under “Comparison with SOTA outlier detection methods”* we mention that SOTA model compression techniques (Frankle & Carbin (2019)) do not deal with processing images from the input stream with varying prior class probabilities. Hence, direct comparison with these methods is not meaningful since their objectives are different from those of ENVISE. *R3 also acknowledges in their strength that our “Experimental evaluation is very thorough”.*
>
> We thank you for taking the time to read our response. We will be happy to answer your questions if you think that they have not been solved.
>
> [1] Zhu, X. and Gong, S.. Knowledge distillation by on-the-fly native ensemble. NeurIPS, 2018.
>
> [2] Chen, D., Mei, J.P., Wang, C., Feng, Y. and Chen, C.. Online Knowledge Distillation with Diverse Peers. AAAI, 2020.

---

### Decision · Program_Chairs · 2021-01-07
**Final Decision**

**Decision:**

Reject

**Comment:**

This paper proposes an online distillation method for efficient object recognition. The main idea is to employ a binary student network to learn frequently occurring classes using the pseudo-labels generated by a teacher network. In order to identify rare vs frequent classes, an attention triplet loss is used. The proposed scheme is empirically evaluated on CIFAR-100 and tiny-imagenet datasets.

The major and common concern from reviewers about this draft is the quality of presentation, which has made it difficult to read and understand the ideas and their underlying motivations. While specific instances of this were mentioned by the reviewers, and responded by the authors, the readability issues go beyond these instances. In fact, I could independently observe presentation issues different from those mentioned by the reviewers. For example, in Eq (1), what is eta (never defined before), what is the loss function here for which the gradient update leads to Eq (1)? The proof also is not clear and seems to have issues. For example, in Eq (13) what is the meaning of multiplying two vectors? If it means dot product, it should be denoted more clearly, e.g. as w^T w^* or <w,w^*>. In Eq (12), can alpha be negative? If not, should be clarified why, and if it can be negative, then (<w^*,w>)^2 >= (alpha n)^2 does not need to hold, but this inequality is used in Eq (14) anyway.

Overall, I think the submission can benefit from an overhaul of the writing. I encourage authors to resubmit after improving on that.